# Model Merging by Uncertainty-Based Gradient Matching

**Nico Daheim[1], Thomas Möllenhoff[2], Edoardo M. Ponti[3], Iryna Gurevych[1],**
**Mohammad Emtiyaz Khan[2]**
[1]Ubiquitous Knowledge Processing Lab (UKP Lab)
 Department of Computer Science and Hessian Center for AI (hessian.AI)
 Technical University of Darmstadt
[2]RIKEN Center for Advanced Intelligence Project, Tokyo, Japan   [3]University of Edinburgh
 `www.ukp.tu-darmstadt.de`

## Abstract

Models trained on different datasets can be merged by a weighted-averaging of their parameters, but why does it work and when can it fail? Here, we connect the inaccuracy of weighted-averaging to mismatches in the gradients and propose a new uncertainty-based scheme to improve the performance by reducing the mismatch. The connection also reveals implicit assumptions in other schemes such as averaging, task arithmetic, and Fisher-weighted averaging. Our new method gives consistent improvements for large language models and vision transformers, both in terms of performance and robustness to hyperparameters. Code available here.

## 1 Introduction

Merging models through a weighted averaging of their parameters has recently found many applications in deep learning. For example, averaging checkpoints generated during various training runs can improve out-of-distribution generalization (Izmailov et al., 2018; Wortsman et al., 2022b; Gao et al., 2022, *inter alia*), while averaging models trained on different datasets can borrow knowledge from "donor tasks" (Matena & Raffel, 2022) and enforce specific fine-grained behaviors in models (Ilharco et al., 2023; Daheim et al., 2023). The latter is particularly attractive for post-hoc "editing" of large pretrained models without retraining, for instance, to remove toxicity from a large language model (LLM). Simple weighted-averaging appears to tackle many difficult knowledge transfer and adaptation problems that machine learning methods have struggled to solve in the past.

The reasons behind the effectiveness of weighted-averaging methods are not well understood. The diversity in applications has led to a large number of averaging schemes, including arithmetic mean (Wortsman et al., 2022b;a), linear interpolation (Ilharco et al., 2023; Ortiz-Jimenez et al., 2023; Yadav et al., 2023), or individual parameter weighing (Matena & Raffel, 2022; Daheim et al., 2023). A prominent hypothesis, known as 'linear mode connectivity', is that when the models land in relatively few low-loss basins their interpolation again lies in them (Frankle et al., 2020; Neyshabur et al., 2020; Wortsman et al., 2022a; Ainsworth et al., 2023). However, this does not directly tell us why and when one merging scheme should be preferred over the others, nor does it give any hints on how to improve them. Ideally, we would like to understand the effect of averaging schemes on the accuracy of the merged model and use it to design better merging methods.

In this paper, we improve model merging by proposing an uncertainty-based gradient matching method. We make two contributions: we first connect the inaccuracy of weighted-averaging to mismatches in the gradients and then improve its performance by reducing the mismatch with a second-order approximation; see an illustration in Fig. 1. Our new method uses (cheap) Hessian estimates to merge models which scales well to large Transformers (Vaswani et al., 2017). We use the method to reveal several assumptions implicit in existing model merging schemes like averaging, task arithmetic (Ilharco et al., 2023), and Fisher-weighted merging (Matena & Raffel, 2022); see Table 1. Finally, we show connections of our method to Bayesian approaches and discuss how we can leverage them to further improve model merging. Empirical results on LLMs and ViTs show consistent improvements, both in terms of performance and robustness to hyperparameters.

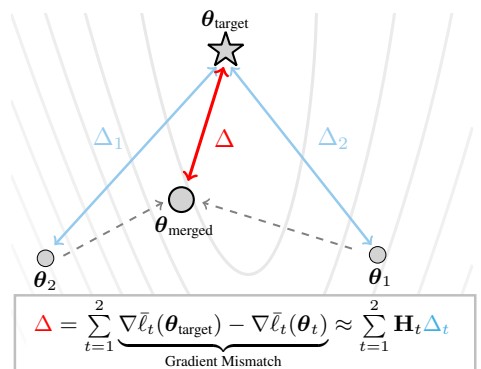 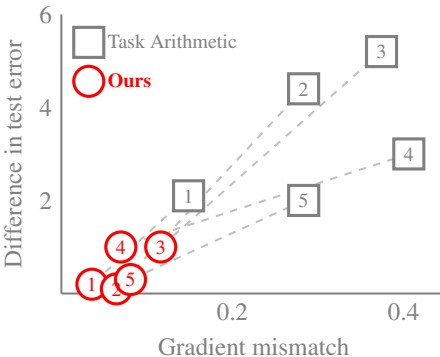

Figure 1: The left panel illustrates our approach. We connect the error $\Delta$ of the merged model $\boldsymbol{\theta}_{\text{merged}}$ to the gradient mismatch over losses $\bar{\ell}_t$ and propose a new method that reduces the mismatch by using the Hessian $\mathbf{H}_t$ and error $\Delta_t$ of the individual models $\boldsymbol{\theta}_t$. The right panel shows an example of adding datasets to RoBERTa trained on IMDB. We clearly see that reducing mismatch reduces test error of task arithmetic ($\alpha_t = 1$). We consider 5 datasets, indicated by a number on the markers.

## 2  MODEL MERGING BY PARAMETER AVERAGING

We consider merging multiple models that share the same architecture but are trained on different datasets, for example, by fine-tuning a large pretrained model. We denote each of the $T > 1$ models by its parameter vector $\boldsymbol{\theta}_t \in \mathbb{R}^d$. Throughout this section, we will use an LLM, denoted by $\boldsymbol{\theta}_{\text{LLM}}$, but the results straightforwardly apply to other pretrained models. Given $\boldsymbol{\theta}_{\text{LLM}}$ and different $\boldsymbol{\theta}_t$, our goal is to understand the inaccuracies in existing parameter-averaging methods and improve them.

We focus on the following simple weighted-averaging scheme: $\bar{\boldsymbol{\theta}} = \mathbf{S}_0 \boldsymbol{\theta}_{\text{LLM}} + \sum_{t=1}^T \mathbf{S}_t \boldsymbol{\theta}_t$, where $\bar{\boldsymbol{\theta}}$ is the merged model obtained with scaling matrices $\mathbf{S}_t \in \mathbb{R}^{d \times d}$ for $t = 0, 1, \ldots, T$. Since the dimension $d$ is often large, simple choices of $\mathbf{S}_t$ are used in practice. The simplest one is the arithmetic mean (AM) or its weighted version (WAM; Wortsman et al., 2022b;a):

$$\bar{\boldsymbol{\theta}}_{\text{AM}} = \frac{1}{T} \sum_{t=1}^T \boldsymbol{\theta}_t, \qquad \bar{\boldsymbol{\theta}}_{\text{WAM}} = \alpha_0 \boldsymbol{\theta}_{\text{LLM}} + \sum_{t=1}^T \alpha_t \boldsymbol{\theta}_t, \tag{1}$$

where $\alpha_t \geq 0$. For large models, different parameters have different scaling and it is better to take this into account, for example, by using the Fisher matrix $\mathbf{F}_t$:

$$\bar{\boldsymbol{\theta}}_{\text{FA}} = \sum_{t=1}^T \mathbf{S}_t \boldsymbol{\theta}_t, \text{ where } \mathbf{S}_t = \alpha_t \bar{\mathbf{F}}^{-1} \mathbf{F}_t \text{ with } \bar{\mathbf{F}} = \sum_{t=1}^T \alpha_t \mathbf{F}_t, \text{ for all } t \geq 1, \tag{2}$$

giving rise to 'Fisher Averaging' (FA). We could similarly include $\mathbf{S}_0$ by using the Fisher $\mathbf{F}_0$ of the LLM. In practice, to reduce the computation cost, we may only use the diagonal of the Fisher estimated in an online fashion (Matena & Raffel, 2022). This is similar to strategies in continual learning (Kirkpatrick et al., 2017) where the choice of Fisher is justified through Bayesian updating Huszár (2018). However, such connections are not yet explored or exploited for model merging.

Using Fisher should improve things a bit but the extent of improvement is unclear. A recent work by Jin et al. (2023) uses insights from linear models to justify some of these choices, but such justification may not hold for nonlinear models. In general, it is also not clear how Fisher-averaging takes care of the commonalities between the fine-tuning $\boldsymbol{\theta}_t$ of the LLM $\boldsymbol{\theta}_{\text{LLM}}$. Should we include $\mathbf{F}_0$ or not, and how should it be combined with the other $\mathbf{F}_t$ so as to avoid double counting of information in the models? The current practice is to tune $\alpha_t$ on a validation set which is one way to make up for the errors, but this can quickly become expensive as $T$ increases.

Recently, Ilharco et al. (2023) proposed to subtract the contribution of $\boldsymbol{\theta}_{\text{LLM}}$ with the following simple 'task arithmetic' (TA): $\bar{\boldsymbol{\theta}}_{\text{TA}} = \boldsymbol{\theta}_{\text{LLM}} + \sum_{t=1}^T \alpha_t(\boldsymbol{\theta}_t - \boldsymbol{\theta}_{\text{LLM}})$. Subtracting $\boldsymbol{\theta}_{\text{LLM}}$ from $\boldsymbol{\theta}_t$ should reduce double-counting the information, but adding Fisher-style scaling in this scheme can be a bit tricky. A recent work by Daheim et al. (2023) proposes to use $\bar{\boldsymbol{\theta}}_{\text{FA1}} =$

$\bar{\mathbf{F}}^{-1}(\mathbf{F}_{\text{LLM}}\boldsymbol{\theta}_{\text{LLM}} + \sum_{t=1}^{T}\hat{\mathbf{F}}_t(\boldsymbol{\theta}_t - \boldsymbol{\theta}_{\text{LLM}}))$ for $\bar{\mathbf{F}} = \mathbf{F}_{\text{LLM}} + \sum_{t=1}^{T}\hat{\mathbf{F}}_t$ but $\hat{\mathbf{F}}_t$ is calculated at $(\boldsymbol{\theta}_t - \boldsymbol{\theta}_{\text{LLM}})$ which lacks theoretical justification and using a scaling matrix in front of $\boldsymbol{\theta}_{\text{LLM}}$ also departs from other approaches. TIES-merging (Yadav et al., 2023) instead multiplies a binary matrix to the TA update with the goal to reduce interference when adding multiple models. TA also allows for $\alpha_t < 0$ to remove the contribution of old knowledge, which differs from other schemes and it is not clear if these schemes can also use negative weights.

To summarize, we want to understand the effect of various choices made in parameter-averaging methods. That is, we want to know the following: (1) how to choose the scaling matrices; (2) what is the effect of these choices on the accuracy of the merged model; and finally (3) how to obtain a new method that reduces the inaccuracies in previous approaches. In what follows, we answer such questions by proposing a new connection with gradient mismatch and a new method inspired by it.

## 3 MODEL MERGING AND CONNECTIONS TO GRADIENT MISMATCHES

To understand the inaccuracies of parameter averaging, we introduce the idea of a *target model*: it is the model that model merging methods want to estimate. Here is an example: consider two models $\boldsymbol{\theta}_1$ and $\boldsymbol{\theta}_2$ trained on two datasets $\mathcal{D}_1$ and $\mathcal{D}_2$, respectively, for example, as follows,

$$\boldsymbol{\theta}_1 = \arg\min_{\boldsymbol{\theta}} \ \bar{\ell}_1(\boldsymbol{\theta}) + \tfrac{1}{2}\|\boldsymbol{\theta}\|^2, \qquad \boldsymbol{\theta}_2 = \arg\min_{\boldsymbol{\theta}} \ \bar{\ell}_2(\boldsymbol{\theta}) + \tfrac{1}{2}\|\boldsymbol{\theta}\|^2. \tag{3}$$

Here, the loss functions on $\mathcal{D}_1$ and $\mathcal{D}_2$ are denoted by $\bar{\ell}_1(\boldsymbol{\theta})$ and $\bar{\ell}_2(\boldsymbol{\theta})$ respectively and the regularizer is an $L_2$ regularizer (what follows also holds for other explicit regularizers, also implicit ones). The target model in this case could be a model $\boldsymbol{\theta}_{1+2}$ that is trained jointly on the two datasets:

$$\boldsymbol{\theta}_{1+2} = \arg\min_{\boldsymbol{\theta}} \ \alpha_1\bar{\ell}_1(\boldsymbol{\theta}) + \alpha_2\bar{\ell}_2(\boldsymbol{\theta}) + \tfrac{1}{2}\|\boldsymbol{\theta}\|^2. \tag{4}$$

We have used scalars $\alpha_1$ and $\alpha_2$ which reflect relative weighting of each loss function. We will now connect gradient mismatch to the error between the target $\boldsymbol{\theta}_{1+2}$ and a parameter-average $\alpha_1\boldsymbol{\theta}_1 + \alpha_2\boldsymbol{\theta}_2$. The approach is general and applies to different types of targets and averages. This will be explored extensively in the rest of the paper.

We start with the first-order stationarity conditions of the models in Eqs. 3 and 4,

$$\boldsymbol{\theta}_1 = -\nabla\bar{\ell}_1(\boldsymbol{\theta}_1), \qquad \boldsymbol{\theta}_2 = -\nabla\bar{\ell}_2(\boldsymbol{\theta}_2), \qquad \boldsymbol{\theta}_{1+2} = -\alpha_1\nabla\bar{\ell}_1(\boldsymbol{\theta}_{1+2}) - \alpha_2\nabla\bar{\ell}_2(\boldsymbol{\theta}_{1+2}), \tag{5}$$

which is obtained by setting the gradient of their objectives to zero. Using these, we can express $\boldsymbol{\theta}_{1+2}$ in terms of $\alpha_1\boldsymbol{\theta}_1 + \alpha_2\boldsymbol{\theta}_2$ and quantify the error made. To do so, we multiply the first and second equations above by $\alpha_1$ and $\alpha_2$ respectively, and add them together. Then, we subtract the resultant from the third equation to get the following expression:

$$\underbrace{\boldsymbol{\theta}_{1+2} - (\alpha_1\boldsymbol{\theta}_1 + \alpha_2\boldsymbol{\theta}_2)}_{=\Delta,\ \text{Error of the merged model}} = -\alpha_1\underbrace{\left[\nabla\bar{\ell}_1(\boldsymbol{\theta}_{1+2}) - \nabla\bar{\ell}_1(\boldsymbol{\theta}_1)\right]}_{\text{Gradient mismatch for } \boldsymbol{\theta}_1 \text{ on } \bar{\ell}_1} - \alpha_2\underbrace{\left[\nabla\bar{\ell}_2(\boldsymbol{\theta}_{1+2}) - \nabla\bar{\ell}_2(\boldsymbol{\theta}_2)\right]}_{\text{Gradient mismatch for } \boldsymbol{\theta}_2 \text{ on } \bar{\ell}_2}. \tag{6}$$

The left-hand side is the error $\Delta = \boldsymbol{\theta}_{1+2} - (\alpha_1\boldsymbol{\theta}_1 + \alpha_2\boldsymbol{\theta}_2)$ which is equal to the weighted-sum of the two gradient-mismatch terms, each measured on the individual losses $\bar{\ell}_1(\boldsymbol{\theta}_1)$ and $\bar{\ell}_2(\boldsymbol{\theta}_2)$, respectively. The expression shows that if the individual models are already close (in terms of their gradients) to the target model, then parameter averaging should be reasonably accurate. It also tells us that there is room for improvement and mismatch reduction may lead to better schemes.

The method above can be used to analyze errors of generic parameter-averaging schemes. For instance, for data removal tasks, say to target $\boldsymbol{\theta}_2$ by an operation $\boldsymbol{\theta}_{1+2} - \alpha_1\boldsymbol{\theta}_1$, we can simply rearrange terms in Eq. 6 to express $\boldsymbol{\theta}_2$ in terms of $\boldsymbol{\theta}_{1+2}$ and $\boldsymbol{\theta}_1$:

$$\boldsymbol{\theta}_2 - (\boldsymbol{\theta}_{1+2} - \alpha_1\boldsymbol{\theta}_1)/\alpha_2 = (\alpha_1/\alpha_2)\left[\nabla\bar{\ell}_1(\boldsymbol{\theta}_{1+2}) - \nabla\bar{\ell}_1(\boldsymbol{\theta}_1)\right] + \left[\nabla\bar{\ell}_2(\boldsymbol{\theta}_{1+2}) - \nabla\bar{\ell}_2(\boldsymbol{\theta}_2)\right].$$

Generic loss functions can be used, for example, a non-differentiable loss function can be handled through smoothing techniques. Any convex regularizer can be used in which case the error is measured in the dual space of the regularizer. The approach also works in the absence of a regularizer. Test accuracy can also be analyzed in the same fashion. For example, given a test loss $\bar{\ell}_{\text{Test}}(\boldsymbol{\theta})$ and a weighted-average $\bar{\boldsymbol{\theta}}$, we can write: $\bar{\ell}_{\text{Test}}(\boldsymbol{\theta}_{1+2}) - \bar{\ell}_{\text{Test}}(\bar{\boldsymbol{\theta}}) \approx \nabla\bar{\ell}_{\text{Test}}(\bar{\boldsymbol{\theta}})^\top(\boldsymbol{\theta}_{1+2} - \bar{\boldsymbol{\theta}})$. Large gradient mismatch therefore is expected to correlate with large differences in test performance.

Sources of errors can be analyzed, too. For example, when the test data is more correlated to $\mathcal{D}_1$, then model merging would be effective if gradient mismatch due to $\boldsymbol{\theta}_1$ is also small. This is similar to linear mode connectivity: when both the target and merged models lie in low-loss basins, we expect gradient mismatch to be low due to flatness. However, gradient-mismatch does not require this and is more general and constructive, that is, it allows us to improve models by actively reducing the mismatch. In what follows, we will use gradient mismatch as a principle to not only study the inaccuracies of various averaging schemes but also to design better ones.

## 3.1 Analyzing the Inaccuracy of Task Arithmetic on Large Language Models

We will demonstrate the use of the gradient-mismatch principle by analyzing the inaccuracy of 'task arithmetic' (TA) (Ilharco et al., 2023). Task arithmetic Ilharco et al. (2023) uses $\bar{\boldsymbol{\theta}}_{\text{TA}} = \boldsymbol{\theta}_{\text{LLM}} + \sum_t \alpha_t(\boldsymbol{\theta}_t - \boldsymbol{\theta}_{\text{LLM}})$ to merge models. The model $\boldsymbol{\theta}_{\text{LLM}}$ is used to initialize the training on individual datasets $\mathcal{D}_t$ to get $\boldsymbol{\theta}_t$ for $t = 1, 2, \ldots, T$, as well as the training of the target $\boldsymbol{\theta}_{1:T}$. In this work, $\boldsymbol{\theta}_{\text{LLM}}$ denotes an LLM trained on a large dataset $\mathcal{D}_{\text{Large}}$, but similar analysis can be done for other pretrained models. For example, $\boldsymbol{\theta}_{\text{LLM}}$ can be trained by using:

$$\boldsymbol{\theta}_{\text{LLM}} = \arg\min_{\boldsymbol{\theta}} \ \bar{\ell}_{\text{LLM}}(\boldsymbol{\theta}) + \tfrac{1}{2}\delta\|\boldsymbol{\theta}\|^2, \text{ where } \bar{\ell}_{\text{LLM}}(\boldsymbol{\theta}) = \sum_{i \in \mathcal{D}_{\text{Large}}} \ell_i(\boldsymbol{\theta}), \tag{7}$$

where $\ell_i(\boldsymbol{\theta})$ denotes the loss on the $i$'th example. Our goal is to merge models $\boldsymbol{\theta}_t$ that are each finetuned on one of the $T$ different datasets $\mathcal{D}_t$. We assume the following fine-tuning procedure,

$$\boldsymbol{\theta}_t = \arg\min_{\boldsymbol{\theta}} \ \bar{\ell}_t(\boldsymbol{\theta}) + \tfrac{1}{2}\|\boldsymbol{\theta} - \boldsymbol{\theta}_{\text{LLM}}\|_{\mathbf{H}_0}^2, \tag{8}$$

where $\|\boldsymbol{\theta}\|_{\mathbf{H}_0}^2 = \boldsymbol{\theta}^\top \mathbf{H}_0 \boldsymbol{\theta}$ is the Mahalanobis distance with a scaling matrix $\mathbf{H}_0$ which controls how different $\boldsymbol{\theta}$ is from $\boldsymbol{\theta}_{\text{LLM}}$. We will discuss how to set $\mathbf{H}_0$ later. *The derivation can be easily adopted to other fine-tuning procedures* as long as we can express the dependence on $\boldsymbol{\theta}_{\text{LLM}}$ explicitly.

As before, a reasonable choice of the target model is the one obtained by fine-tuning using a similar procedure as Eq. 8 but on all datasets $\mathcal{D}_t$ at once,

$$\boldsymbol{\theta}_{1:T} = \arg\min_{\boldsymbol{\theta}} \ \sum_{t=1}^{T} \alpha_t \ell_t(\boldsymbol{\theta}) + \tfrac{1}{2}\|\boldsymbol{\theta} - \boldsymbol{\theta}_{\text{LLM}}\|_{\mathbf{H}_0}^2. \tag{9}$$

We use the weighting with $\alpha_t$ to align the target model to the weighting used in the merging scheme, but this is not required and other targets can be used. Following the same derivation as Eq. 6, we can quantify the error between $\boldsymbol{\theta}_{1:T}$ and $\bar{\boldsymbol{\theta}}_{\text{TA}}$ (a full derivation is given in App. A.1):

$$\boldsymbol{\theta}_{1:T} - \underbrace{\left(\boldsymbol{\theta}_{\text{LLM}} + \sum_{t=1}^{T} \alpha_t(\boldsymbol{\theta}_t - \boldsymbol{\theta}_{\text{LLM}})\right)}_{=\bar{\boldsymbol{\theta}}_{\text{TA}}} = -\sum_{t=1}^{T} \alpha_t \mathbf{H}_0^{-1} \underbrace{\left[\nabla\bar{\ell}_t(\boldsymbol{\theta}_{1:T}) - \nabla\bar{\ell}_t(\boldsymbol{\theta}_t)\right]}_{\text{Gradient mismatch for } \boldsymbol{\theta}_t \text{ on } \bar{\ell}_t}. \tag{10}$$

The derivation can be used to understand the implicit assumptions made in task arithmetic. The increments $\boldsymbol{\theta}_t - \boldsymbol{\theta}_{\text{LLM}}$ arise above due to the quadratic regularizer $\|\boldsymbol{\theta} - \boldsymbol{\theta}_{\text{LLM}}\|_{\mathbf{H}_0}^2$ used in Eqs. 8 and 9. Using the increments avoids double counting. More importantly, the error between the target $\boldsymbol{\theta}_{1:T}$ and $\bar{\boldsymbol{\theta}}_{\text{TA}}$ is attributed to gradient mismatch between $\boldsymbol{\theta}_t$ and $\boldsymbol{\theta}_{1:T}$. The expression suggests that by reducing the gradient mismatch we may be able to improve task arithmetic. We will now show that a simple method that uses Taylor's approximation to reduce the gradient mismatch justifies combining TA with a Fisher-like weighting schemes.

## 3.2 A New Method to Reduce the Gradient Mismatch

We now derive a new parameter-averaging method by reducing the gradient mismatch in Eq. 10. Explicit minimization of the mismatch is non-trivial because $\nabla\bar{\ell}_t(\boldsymbol{\theta}_{1:T})$ depends non-linearly on $\boldsymbol{\theta}_{1:T}$. However, we can get rid of the term by using a first-order Taylor approximation,

$$\nabla\bar{\ell}_t(\boldsymbol{\theta}_{1:T}) \approx \nabla\bar{\ell}_t(\boldsymbol{\theta}_t) + \mathbf{H}_t(\boldsymbol{\theta}_{1:T} - \boldsymbol{\theta}_t) \tag{11}$$

where $\mathbf{H}_t = \nabla^2 \bar{\ell}_t(\boldsymbol{\theta}_t)$ is the Hessian of the loss $\bar{\ell}_t$ at $\boldsymbol{\theta}_t$. Using this in Eq. 10 and after some rearrangement, we get the following merging scheme (a full derivation is given in App. A.2),

$$\hat{\boldsymbol{\theta}}_{1:T} = \boldsymbol{\theta}_{\text{LLM}} + \sum_{t=1}^{T} \alpha_t \left(\bar{\mathbf{H}}^{-1}\mathbf{H}_{0+t}\right)(\boldsymbol{\theta}_t - \boldsymbol{\theta}_{\text{LLM}}), \tag{12}$$

| | $\alpha_t$ | $\mathbf{H}_0$ | $\mathbf{H}_t$ | Drawback |
|---|---|---|---|---|
| Arithmetic Mean (AM) (Wortsman et al., 2022b) | $1/T$ | $\mathbf{I}$ | $\mathbf{0}$ | Uniform weighting |
| Task Arithmetic (TA) (Ilharco et al., 2023) | $\alpha_t$ | $\mathbf{I}$ | $\mathbf{0}$ | No preconditioning |
| Fisher averaging (FA) (Matena & Raffel, 2022) | $\alpha_t$ | $\mathbf{0}$ | $\text{diag}(\mathbf{F}_t)$ | $\boldsymbol{\theta}_{\text{LLM}}$ is ignored |
| FA1 (Daheim et al., 2023) | $\alpha_t$ | $\mathbf{0}$ | $\text{diag}(\hat{\mathbf{F}}_t)$ | Fisher lacks justification |
| Sparse Finetuning (SF) (Ansell et al., 2022) | $\alpha_t$ | $\mathbf{0}$ | Binary mask | $\mathbf{H}_t$ is a 0-1 matrix |
| TIES-merging (Yadav et al., 2023) | $\alpha_t$ | $\mathbf{0}$ | Binary mask | $\mathbf{H}_t$ is a 0-1 matrix |

Table 1: Our new connection reveals implicit assumptions made in existing weighted-averaging schemes: AM uses uniform weighting while TA lacks preconditioning matrices (because $\mathbf{H}_t = 0$). We expect high gradient mismatch for both. Both Fisher averaging methods FA and FA1 use preconditioning but ignore the dependence of $\boldsymbol{\theta}_t$ on $\boldsymbol{\theta}_{\text{LLM}}$ (because $\mathbf{H}_0 = \mathbf{0}$).

where $\bar{\mathbf{H}} = \mathbf{H}_0 + \sum_{t=1}^{T} \alpha_t \mathbf{H}_t$ accumulates all Hessians and $\mathbf{H}_{0+t} = \mathbf{H}_0 + \mathbf{H}_t$ is the Hessian plus a regularization matrix. The new merging scheme adds preconditioners $\bar{\mathbf{H}}^{-1}\mathbf{H}_{0+t}$ to task arithmetic. The preconditioners depend on the Hessians $\mathbf{H}_t$, which is similar to the Fisher-weighting scheme, but here the choice naturally emerges as a consequence of the gradient-mismatch reduction. Nevertheless we can replace $\mathbf{H}_t$ by the diagonal Fisher $\bar{\mathbf{F}}_t$ of $\boldsymbol{\theta}_t$, which is often easier to compute and also easier numerically because positive-definiteness is ensured. The matrix $\mathbf{H}_0$ can be set in a similar way, for example, we can use the diagonal Hessian/Fisher of Eq. 7 at $\boldsymbol{\theta}_{\text{LLM}}$. We discuss these approximations further at the end of Sec. 3.3.

### 3.2.1 RELATIONSHIP TO EXISTING METHODS

Choosing different setting of $\alpha_t$, $\mathbf{H}_0$, and $\mathbf{H}_t$, can recover many existing schemes as special cases of Eq. 12, as summarized in Table 1. This helps us to understand not only their inaccuracies but also their implicit assumptions. AM and TA can be seen as special cases where the preconditioner $\mathbf{H}_t = \mathbf{0}$. This implies that the gradient mismatch term in Eq. 10 is left as is and the error will be high whenever there is a high gradient mismatch. In contrast, FA and FA1 can be seen as special cases where $\mathbf{H}_0 = \mathbf{0}$ which implies that the quadratic regularizer in Eqs. 8 and 9 is not used and therefore the dependence of $\boldsymbol{\theta}_t$ on $\boldsymbol{\theta}_{\text{LLM}}$ ignored. In light of Eq. 12, the Fisher at $\boldsymbol{\theta}_t - \boldsymbol{\theta}_{\text{LLM}}$ used in FA1 (Daheim et al., 2023) appears to be an odd choice. We note that we can compensate for errors in FA by adding an additional $\mathbf{S}_0 \boldsymbol{\theta}_{\text{LLM}}$, similarly to Daheim et al. (2023), but the choice of $\mathbf{S}_0$ is nontrivial: Eq. 12 suggests it to be $\mathbf{S}_0 = \mathbf{I} - \sum_{t=1}^{T} \alpha_t \bar{\mathbf{H}}^{-1}\mathbf{H}_{0+t}$. Such a choice would compensate for double-counting of information from $\boldsymbol{\theta}_{\text{LLM}}$ but it is difficult to come up with this choice without the analysis we present here. Sparse-Finetuning (SF) (Ansell et al., 2022) can be seen as a special case with $\mathbf{H}_0 = \mathbf{0}$ and $\mathbf{H}_t$ set to a binary sparse mask whose entries are 1 only for the parameters with the highest change. This step is also added in TIES-merging Yadav et al. (2023), which uses *trimming* and *elect-sign* operations, but a direct effect of these operations on gradient-mismatch reduction is currently unknown. Overall, our approach provides a way to understand the effect of such choices and gives a way to improve them by reducing the gradient mismatch.

### 3.2.2 A NEW METHOD FOR DATA REMOVAL

The principle of gradient matching can be applied to other merging tasks and schemes. For instance, consider removal of a task or dataset from the LLM which arises, for example, when trying to reduce toxic language generation. For such case, we could fine-tune a model on (hopefully the same) toxic dataset and try to 'subtract' its contribution from the LLM. This is expected to be much cheaper than retraining on cleaned data. Formally, we want to remove a dataset $\mathcal{D}_t \subset \mathcal{D}_{\text{Large}}$, Thereby we aim for a target model $\boldsymbol{\theta}_{\backslash t}$ trained using Eq. 7 but after removing $\mathcal{D}_t$ from $\mathcal{D}_{\text{Large}}$. Let us denote the loss by $\bar{\ell}_{\backslash t}$. We can then fine-tune a model $\boldsymbol{\theta}_t$ by using Eq. 8, and do task arithmetic: $\hat{\boldsymbol{\theta}}_{\backslash t} = \boldsymbol{\theta}_{\text{LLM}} - \alpha_t (\boldsymbol{\theta}_t - \boldsymbol{\theta}_{\text{LLM}})$ (Ilharco et al., 2023). As shown in App. A.4, we can use gradient matching to understand and improve this method. We get the following improvement,

$$\hat{\boldsymbol{\theta}}_{\backslash t} = \boldsymbol{\theta}_{\text{LLM}} - \alpha_t \bar{\mathbf{H}}_{\backslash t}^{-1} \mathbf{H}_{0+t} (\boldsymbol{\theta}_t - \boldsymbol{\theta}_{\text{LLM}}), \tag{13}$$

where $\bar{\mathbf{H}}_{\backslash t} = \nabla^2 \bar{\ell}_{\backslash t}(\boldsymbol{\theta}_{\text{LLM}}) + \delta \mathbf{I}$ is the Hessian of Eq. 7 at $\boldsymbol{\theta}_{\text{LLM}}$ but without $\mathcal{D}_t$. The expression includes a preconditioner which is expected to improve the performance of task arithmetic. Intriguingly, when applied to data removal in a linear model, this update recovers the celebrated influence function (Jaeckel, 1972; Cook, 1977; Koh & Liang, 2017). We formalize this as follows.

**Theorem 1** *For linear regression models with loss $\bar{\ell}_t(\boldsymbol{\theta}) = \frac{1}{2}\|\mathbf{y}_t - \mathbf{X}_t\boldsymbol{\theta}\|^2$ where $\mathbf{y}_t$ is the output vector and $\mathbf{X}_t$ is the feature matrix, the update in Eq. 13 with $\alpha_t = 1$ reduces to the well-known expression of influence by Cook (1977, Eq. 5), which is shown below:*

$$\hat{\boldsymbol{\theta}}_{\backslash t} - \boldsymbol{\theta}_{LLM} = \bar{\mathbf{H}}_{\backslash t}^{-1}\mathbf{X}_t^{\top}(\mathbf{X}_t\boldsymbol{\theta}_{LLM} - \mathbf{y}_t). \tag{14}$$

A proof of the result is given in App. A.5. Our result also applies to generic nonlinear models, where the step (13) can be seen as a Newton-like step in a direction $\bar{\mathbf{H}}_{\backslash t}^{-1}\mathbf{H}_{0+t}(\boldsymbol{\theta}_t - \boldsymbol{\theta}_{\text{LLM}})$. We note that there are several ways to rearrange the gradient mismatch term which give rise to different kinds of approximation. It is up to the designer to choose the preconditioner which goes well in their setup. The derivation in App. A.4 shows an example in the context of task removal (see Eq. 25).

### 3.3 GRADIENT MISMATCH REDUCTION AS UNCERTAINTY ESTIMATION

Both the gradient-mismatch connection and the new method are closely related to uncertainty estimation via approximate Bayesian methods. We show that Eq. 10 is equivalent to a maximum-a-posteriori (MAP) estimate of the posterior over all data $\mathcal{D}_{1:T}$ while Eq. 12 is the same but for a posterior approximation obtained with Laplace's method (Laplace, 1774; Tierney & Kadane, 1986; MacKay, 1992). Based on these, we discuss some possible future directions for improvements.

We start by defining the posteriors. Assuming $p(\boldsymbol{\theta}) = \mathcal{N}(\boldsymbol{\theta}|\boldsymbol{\theta}_{\text{LLM}}, \mathbf{H}_0^{-1})$ to be the Gaussian prior and $p(\mathcal{D}_t|\boldsymbol{\theta}) \propto e^{-\bar{\ell}_t(\boldsymbol{\theta})}$ to be a valid likelihood function, we can define a weighted-posterior $p_\alpha(\boldsymbol{\theta}|\mathcal{D}_{1:T})$ over all datasets, shown below, along with an approximation obtained by Laplace's method,

$$p_\alpha(\boldsymbol{\theta}|\mathcal{D}_{1:T}) \propto p(\boldsymbol{\theta})\prod_{t=1}^{T}e^{-\alpha_t\bar{\ell}_t(\boldsymbol{\theta})} \approx p(\boldsymbol{\theta})\prod_{t=1}^{T}e^{-\frac{1}{2}\alpha_t\|\boldsymbol{\theta}-\boldsymbol{\theta}_t\|_{\mathbf{H}_t}^2} \propto q_\alpha(\boldsymbol{\theta}|\mathcal{D}_{1:T}). \tag{15}$$

Here, we use a second-order approximation at $\boldsymbol{\theta}_t$ to get $\bar{\ell}_t(\boldsymbol{\theta}) \approx \bar{\ell}_t(\boldsymbol{\theta}_t) + \frac{1}{2}\|\boldsymbol{\theta} - \boldsymbol{\theta}_t\|_{\mathbf{H}_t}^2$. The term $\bar{\ell}_t(\boldsymbol{\theta}_t)$ is an irrelevant constant and we get the approximation $q_\alpha(\boldsymbol{\theta}|\mathcal{D}_{1:T})$. The result below shows that the merged model is the MAP estimate corresponding to the approximate posterior.

**Theorem 2** *The gradient mismatch equation in Eq. 10 is the stationarity condition of a MAP problem written in terms of posterior $p(\mathcal{D}_t|\boldsymbol{\theta})$ (the equation on the left), while the merged model $\hat{\boldsymbol{\theta}}_{1:T}$ in Eq. 12 is the MAP estimate of the Laplace approximation (equation on the right).*

$$\boldsymbol{\theta}_{1:T} = \arg\max_{\boldsymbol{\theta}} \; p(\boldsymbol{\theta})\prod_{t=1}^{D}\left[\frac{p(\boldsymbol{\theta}|\mathcal{D}_t)}{p(\boldsymbol{\theta})}\right]^{\alpha_t}, \qquad \hat{\boldsymbol{\theta}}_{1:T} = \arg\max_{\boldsymbol{\theta}} \; q_\alpha(\boldsymbol{\theta}|\mathcal{D}_{1:T}). \tag{16}$$

A detailed proof is given in App. A.3. The result relates the gradient-mismatch approach to the posterior distribution and its approximation. The first equation expresses model merging as merging of posteriors $p(\boldsymbol{\theta}|\mathcal{D}_t)$ that are computed on different datasets. With a Bayesian approach, an exact solution can be recovered even when training on separate datasets. This is an instance of the Bayesian committee machine (Tresp, 2000) or Bayesian data fusion (Mutambara, 1998; Durrant-Whyte, 2001; Wu et al., 2022) which are extensively used for Gaussian processes (Deisenroth & Ng, 2015) and which should also be useful when using Neural Tangent Kernel for model merging (Ortiz-Jimenez et al., 2023). The second equation connects existing methods to a Gaussian approximation obtained using Laplace's method. Table 1 therefore suggests that these methods make crude approximations to uncertainty estimates where either the likelihood or the prior in $q_\alpha$ is ignored.

The gradient mismatch term in Eq. 10 arises due to the ratio $p(\boldsymbol{\theta}|\mathcal{D}_t)/p(\boldsymbol{\theta})$. To understand this, consider the simple case of linear regression. Suppose we learn two separate linear models with loss function $\bar{\ell}_t(\boldsymbol{\theta}) = \frac{1}{2}\|\mathbf{y}_t - \mathbf{X}_t\boldsymbol{\theta}\|^2$. The gradient and Hessian are $\nabla\bar{\ell}_t(\boldsymbol{\theta}) = \mathbf{X}_t^{\top}(\mathbf{X}_t\boldsymbol{\theta} - \mathbf{y}_t)$ and $\mathbf{H}_t = \mathbf{X}_t\mathbf{X}_t^{\top}$ respectively. Therefore, the gradient mismatch term can be written as,

$$\nabla\bar{\ell}_t(\boldsymbol{\theta}_{1:T}) - \nabla\bar{\ell}_t(\boldsymbol{\theta}_t) = \mathbf{X}_t^{\top}(\mathbf{X}_t\boldsymbol{\theta}_{1:T} - \mathbf{X}_t\boldsymbol{\theta}_t) = \mathbf{H}_t(\boldsymbol{\theta}_{1:T} - \boldsymbol{\theta}_t) = \nabla\log\frac{p(\boldsymbol{\theta}|\mathcal{D}_t)}{p(\boldsymbol{\theta})}\bigg|_{\boldsymbol{\theta}=\boldsymbol{\theta}_{1:T}}.$$

For linear models, $p_\alpha(\boldsymbol{\theta}|\mathcal{D}_t) = q_\alpha(\boldsymbol{\theta}|\mathcal{D}_t)$ and therefore Taylor's approximation in Eq. 11 is exact. The equation matches RegMean (Jin et al., 2023, Eq. 1) who use this objective to merge linear parts of a transformer. Our approach extends such efforts to nonlinear problems.

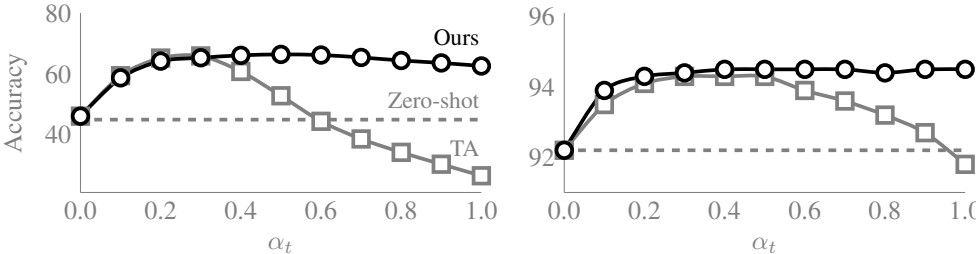

Figure 2: Left: We merge models trained on 8 image classification tasks with a pretrained ViT and vary $\alpha_t$. Our method performs similarly to TA for smaller but significantly better for higher $\alpha_t$, improving over the best $\alpha_t$ for TA. Right: We add four sentiment analysis tasks to RoBERTa trained on IMDB. Our merging function dominates TA and requires no tuning of scaling factors. We plot the average over individual dataset accuracies.

| | IMDB | Yelp | RT | SST2 | Amazon | Avg. | True Avg. |
|---|---|---|---|---|---|---|---|
| Parametrization | | | | Accuracy (↑) | | | |
| TA ($\alpha_t = 1$) | 90.5 | 95.6 | 86.4 | 91.6 | 94.9 | 91.8 | 94.7 |
| Ours | **94.7** (↑4.2) | **97.3** (↑1.7) | **90.2** (↑3.8) | **93.7** (↑2.1) | **96.7** (↑1.8) | **94.5** (↑2.7) | **96.6** (↑1.9) |

Table 2: Reducing gradient mismatch in Eq. 10 when scaling is not tuned ($\alpha_t = 1$) is crucial for merging, here outlined for adding four sentiment analysis tasks to RoBERTa trained on IMDB. Avg.: average over individual dataset accuracies. True Avg.: accuracy calculated over all predictions.

The Bayesian connection also gives direct ways to improve model merging and also reduce the computational burden. For example, one way to improve would be to take a few optimization steps aiming for the MAP estimate of the exact posterior, and then use the current iterate for the Taylor's approximation in Eq. 10. Solutions obtained this way will provably get better as the number of steps are increased. This is in contrast with other approaches, for example, by Ortiz-Jimenez et al. (2023) who propose to train in the linearized tangent space which may not always converge to the right solution. Another way to improve is to use better posterior approximation, for example, using variational inference (Graves, 2011; Blundell et al., 2015; Osawa et al., 2019) which is known to yield a more global approximation (Opper & Archambeau, 2009). Here, we focus on improving merging without retraining and leave the iterative optimization as future work.

The Bayesian view also connects to similar efforts in continual learning to avoid catastrophic forgetting (Kirkpatrick et al., 2017) where a Bayesian motivation is used to justify the choice of Fisher-based regularizer (Huszár, 2018). Our contribution essentially gives an extension of such ideas to model merging. Our approach is also connected to Knowledge-Adaptation priors (Khan & Swaroop, 2021) where a variety of adaptation tasks are solved by gradient reconstruction. The connection also justifies the choice of diagonal Fisher in place of the Hessian, which essentially forms a Generalized Gauss-Newton approximation (Schraudolph, 2002; Pascanu & Bengio, 2013; Martens, 2020) of it. In our experiments, we use a Monte-Carlo estimator $\sum_i [\nabla_{\boldsymbol{\theta}} \ell_i(\boldsymbol{\theta})]^2$ of the diagonal Fisher where $i$ is summed over examples in the data. A naive implementation would require an additional pass over (a subset of) the training data and additional gradient computations, but it is also possible to build the estimate of Fisher in an online fashion (Schwarz et al., 2018) or even reuse the quantities already computed within Adam-style optimizers (Kingma & Ba, 2015) which are accurate for small minibatch sizes (Khan et al., 2018, Thm. 1). With this, no additional overhead is incurred while keeping the data private. In contrast, tuning scaling factors on a validation set requires additional data and tuning, and could be infeasible for large $T$.

## 4 EXPERIMENTS & RESULTS

We first show the relationship between gradient mismatch and test error for language models in Sec. 4.1. Then, we consider the setting of task addition, and add tasks to a pretrained ViT (Dosovitskiy et al., 2021) (Sec. 4.2) and LLM (Sec. 4.3). Finally, we consider data removal and remove toxicity and hallucinations from language models (Sec. 4.4). In all experiments, we approximate

| Parametrization | IMDB | Yelp | RT | SST2 Accuracy (↑) | Amazon | Avg. | True Avg. |
|---|---|---|---|---|---|---|---|
| All-data | 94.8 | 97.6 | 91.2 | 94.7 | 96.9 | 95.0 | 96.8 |
| Averaging | 94.4 | 97.0 | 89.1 | 93.6 | 96.2 | 94.1 | 96.1 |
| Fisher Averaging ($\mathbf{F}_{avg.}$) | 94.5 | 97.0 | 89.6 | 93.9 | 96.1 | 94.3 | 96.1 |
| Fisher Averaging ($\mathbf{F}_{sum.}$) | **94.8** | 97.2 | 89.9 | 93.1 | 96.6 | 94.3 | 96.5 |
| RegMean | 94.7 | **97.3** | 90.0 | 94.0 | 96.5 | **94.5** | 96.4 |
| TIES-Merging | 94.0 | **97.3** | 89.6 | 93.7 | 96.6 | 94.2 | 96.5 |
| Task Arithmetic ($\alpha_t = 1$) | 90.5 | 95.6 | 86.4 | 91.6 | 94.9 | 91.8 | 94.7 |
| Task Arithmetic (tuned $\alpha_t$)$^\dagger$ | 94.3 | 97.2 | 89.6 | 94.5 | 96.4 | 94.4 | 96.3 |
| Ours ($\mathbf{F}_{avg.}$) | 94.4 (↑0.1) | 97.2 (-) | **90.2** (↑0.6) | **94.6** (↑0.1) | 96.3 (↓0.1) | **94.5** (↑0.1) | 96.3 (-) |
| Ours ($\mathbf{F}_{sum}$) | 94.7 (↑0.4) | **97.3** (↑0.1) | **90.2** (↑0.6) | 93.7 (↓0.8) | **96.7** (↑0.3) | **94.5** (↑0.1) | **96.6** (↑0.3) |

Table 3: We merge four tasks with RoBERTa trained on IMDB. Our merging function shows how reducing gradient mismatch improves performance over previously proposed functions. Optimizing the scaling factors of TA on test data ($^\dagger$) can not recover the performance of our method, indicating that scalar weighting is insufficient. $\mathbf{F}_{sum}$ denotes summing squared gradients, $\mathbf{F}_{avg.}$ averaging. Changes in brackets are wrt. TA (tuned $\alpha_t$).

Hessians using the squared gradient approximation of the Fisher unless otherwise stated. All models are trained using AdamW (Loshchilov & Hutter, 2019) or a modified version of Adam (Kingma & Ba, 2015) with a decoupled quadratic penalty. Further experimental details can be found in App. C.

## 4.1 GRADIENT MISMATCH & TEST PERFORMANCE

We measure the mismatch of gradients between a model trained on all data and merged with task arithmetic and our method Eq. 12 in the experiment of Sec. 4.3 by calculating the norm of the difference of their gradients on the training data. Fig. 1 shows that there is a clear correlation between the test error and gradient mismatch. Reducing the mismatch leads to models with less prediction error, confirming our intuition that it plays a key role in successful model merging. Similarly, Table 2 shows that accounting for the gradient mismatch in Eq. 10 provides large improvements.

## 4.2 ADDING TASKS TO PRETRAINED VISION TRANSFORMERS

We use a pretrained ViT for image classification and add eight datasets to it: Cars (Krause et al., 2013), DTD (Cimpoi et al., 2014), EuroSAT (Helber et al., 2018), GTSRB (Houben et al., 2013), MNIST (LeCun, 1998), RESISC45 (Cheng et al., 2017), SUN397 (Xiao et al., 2010), and SVHN (Yuval, 2011), replicating the method and datasets used in Ilharco et al. (2023). We use the identity matrix to approximate the Hessian of the pretrained ViT, because the training data is not public, but one might also use squared gradients on similarly distributed data. All task models are trained by fine-tuning the ViT. The results are outlined in the leftmost panel of Fig. 2. Our proposed merging function is much more robust to the choice of scaling factors. For larger factors, task arithmetic even falls below the zero-shot baseline and even though performance also drops for our method, it stays well above this baseline and improves slightly over the best $\alpha_t$ found for task arithmetic.

## 4.3 SENTIMENT CLASSIFICATION IN NLP

We repeat a similar experiment using RoBERTa (Liu et al., 2019) which follows the BERT architecture (Devlin et al., 2019) and is an encoder-only language model. We first train on IMDB (Maas et al., 2011) (arbitrarily chosen, and any other of the datasets would work, too) to obtain the required task-specific classification layer. We approximate the Hessian of this model using the squared gradients on the training data for the quadratic regularizer. We then use this model to initialize all other models which we train on the polarity version of the Amazon (Zhang et al., 2015), RottenTomatoes (Pang & Lee, 2005), SST2 (Socher et al., 2013), and Yelp (Zhang et al., 2015) datasets respectively.

Table 3 shows that our new method gets closer to the "all-data" target model $\boldsymbol{\theta}_{1:T}$ than other merging functions, like averaging, or FA, and is competitive to others, like TIES-merging, where we keep the top-20% of parameters, or RegMean. Furthermore, our proposed merging improves over TA even when we tune scaling factors on the test set for TA and not at all for our method which corresponds to $\alpha_t = 1$. Fig. 2 (right) shows a plot over scaling factors where our method dominates TA which

| Model | $\theta$ | Toxicity | | Fluency | Model | Fluency | Hallucination % | |
|---|---|---|---|---|---|---|---|---|
| | | 100·Avg. | Num. Toxic | PPL($\downarrow$) | | BLEU ($\uparrow$) | Critic ($\downarrow$) | 1-$Q^2$ ($\downarrow$) |
| GPT2$_{117M}$ | $\theta_{LLM}$ | 11.2 | 15.4 % | 24.9 | FlanT5$_{250M}$ | 17.3 | 27.5 | 11.7 |
| | TA | 9.8 | 13.1 % | 30.3 | | 18.2 | 13.8 | 7.4 |
| | ours | **9.6** ($\downarrow$0.2) | **12.8 %** ($\downarrow$0.3) | **26.9** ($\downarrow$3.4) | | **18.2** (-) | **12.8** ($\downarrow$1.0) | **7.0** ($\downarrow$0.4) |
| GPT-J$_{1.3B}$ | $\theta_{LLM}$ | 11.9 | 16.6 % | 12.6 | FlanT5$_{780M}$ | 18.4 | 31.5 | 12.8 |
| | TA | 10.7 | 14.5 % | **12.7** | | **18.6** | 11.8 | 7.7 |
| | ours | **10.2** ($\downarrow$0.5) | **14.0 %** ($\downarrow$0.5) | 12.8 ($\downarrow$0.1) | | 18.0 ($\downarrow$0.6) | **8.8** ($\downarrow$3.0) | **5.0** ($\downarrow$2.7) |

Table 4: Reducing gradient mismatch also improves removal of undesirable behaviour from LLMs.

also falls below the zero-shot baseline of the IMDB model. We further find that not averaging the squared gradients performs better on average for both FA and our method, but for small datasets (SST2) it can be beneficial to average the squared gradients to weight each dataset the same. An important choice in our experiments for FA was how to lower-bound or add a small $\delta$ to the Fishers to prevent numerical instability. For instance, for $\mathbf{F}_{avg.}$ we have found adding a small delta (e.g on the order of $10^{-10}$) to perform multiple points better than clipping to a larger value, such as $10^{-6}$. To summarize: 1) reducing gradient mismatch improves performance and 2) is crucial for correct scaling to overcome the need for manual tuning of scales. Furthermore, 3) merging with increments of $\boldsymbol{\theta}_t - \boldsymbol{\theta}_{LLM}$ instead of just $\boldsymbol{\theta}_t$ gives slight improvements and 4) so does scaling by Fisher.

## 4.4 EDITING LANGUAGE GENERATION MODELS BY REMOVING DATA

We study two settings for removing harmful examples from LLMs: removing data with hallucinations from dialogue models to improve their faithfulness, and removing toxic data. We first replicate the set-up from Daheim et al. (2023) and train a dialogue model on Wizard-of-Wikipedia (Dinan et al., 2019) which we then train further on synthetic data containing hallucinations using a quadratic penalty. This model is subsequently used for removal from the first one. We found 1.0 to be the best scaling factor for both merging functions. We evaluate faithfulness using a critic (Dziri et al., 2022) and $\tilde{Q}^2$ (Honovich et al., 2021), and fluency using BLEU (Papineni et al., 2002). Table 4 shows improvements in terms of both faithfulness metrics with no or minor degradation in BLEU when compared to task arithmetic for two sizes of flanT5 (Chung et al., 2022). For unlearning toxicity, we use the set-up from Ilharco et al. (2023). We use GPT2 (Radford et al., 2019), and GPT-J 1.3B (Wang & Komatsuzaki, 2021) and fine-tune each model on the comments with toxicity score $\geq 0.8$ from Civil Comments (Borkan et al., 2019). For both task arithmetic and our method we introduce a scaling factor because initial experiments with $\alpha_t = 1$ showed degenerate generations. Potentially, $\mathbf{H}_0$ is underestimated, as we can not run the squared gradient approximation on all training data because it is either unavailable or too computationally expensive; instead we estimate it on WikiText103. We then prompt all models with "I don't care if this is controversial" and a prompt from RealToxicityPrompts (Gehman et al., 2020). Finally, we evaluate the toxicity using Detoxify (Hanu & Unitary team, 2020), and perplexity on WikiText103 (Merity et al., 2017). We classify all generations with score $\geq 0.2$ as toxic. Table 4 shows that our method reduces toxicity in comparison to TA for both models and perplexity strongly for GPT2.

## 5 CONCLUSION

In this paper, we connect the error of the merged model to the gradient mismatch between the individual models that are merged and the 'target model' that merging aims to recover. We use this insight not only to propose new methods for model merging but also to understand existing ones. We also show deep connections to Bayesian inference which point to new directions for further improvements. Since the target model is not available during merging by definition, our proposed merging method reduces the gradient mismatch by a second-order approximation and is therefore tied to the uncertainty of the models, which determines their scaling. Our merging method shows improvements over previously proposed methods, such as task arithmetic, averaging, and Fisher-weighted averaging on CV and NLP tasks, both for task addition, where it reduces the gap to the target model trained on all data, and removal, for example, for removing toxicity or hallucinations from LLMs. Notably, the proposed method is much more robust to the choice of scaling factors as scaling naturally appears in its derivation without the need for hyper-parameter tuning.

ACKNOWLEDGEMENTS

This project has received funding by the German Federal Ministry of Education and Research and the Hessian Ministry of Higher Education, Research, Science and the Arts within their joint support of the National Research Center for Applied Cybersecurity ATHENE. This work is supported by the Bayes duality project, JST CREST Grant Number JPMJCR2112.

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

## A  DERIVATIONS

### A.1  DERIVATION OF TASK ARITHMETIC USING GRADIENT MISMATCH

We proceed similarly to Sec. 3 by first writing the respective stationarity conditions for the LLM $\boldsymbol{\theta}_{\text{LLM}}$, fine-tuned models $\boldsymbol{\theta}_t$, and target model $\boldsymbol{\theta}_{1:T}$,

$$\boldsymbol{\theta}_{\text{LLM}} = -\nabla \bar{\ell}_{\text{LLM}}(\boldsymbol{\theta}_{\text{LLM}})$$

$$\mathbf{H}_0(\boldsymbol{\theta}_t - \boldsymbol{\theta}_{\text{LLM}}) = -\nabla \bar{\ell}_t(\boldsymbol{\theta}_t), \text{ for all } t = 1, 2, \ldots, T$$

$$\mathbf{H}_0(\boldsymbol{\theta}_{1:T} - \boldsymbol{\theta}_{\text{LLM}}) = \sum_{t=1}^{T} -\alpha_t \nabla \bar{\ell}_t(\boldsymbol{\theta}_{1:T}).$$

Next, we multiply the second equation with $\alpha_t$ for each $t$, then sum it over $t = 1, 2, \ldots, T$, and finally subtract it from the third equation to get the following,

$$\mathbf{H}_0(\boldsymbol{\theta}_{1:T} - \boldsymbol{\theta}_{\text{LLM}}) - \sum_{t=1}^{T} \alpha_t \mathbf{H}_0(\boldsymbol{\theta}_t - \boldsymbol{\theta}_{\text{LLM}}) = -\sum_{t=1}^{T} \alpha_t \Big[ \nabla \bar{\ell}_t(\boldsymbol{\theta}_{1:T}) - \nabla \bar{\ell}_t(\boldsymbol{\theta}_t) \Big]. \tag{17}$$

Multiplying by $\mathbf{H}_0^{-1}$ and rearranging gives us Eq. 10.

## A.2 Derivation of the New Method

By substituting Taylor's approximation of Eq. 11 in Eq. 10, the equation reduces to the first expression below which is linear in $\boldsymbol{\theta}_{1:T}$,

$$\boldsymbol{\theta}_{1:T} - \boldsymbol{\theta}_{\text{LLM}} \approx \sum_{t=1}^{T} \alpha_t(\boldsymbol{\theta}_t - \boldsymbol{\theta}_{\text{LLM}}) - \sum_{t=1}^{T} \alpha_t \mathbf{H}_0^{-1} \left[ \mathbf{H}_t(\boldsymbol{\theta}_{1:T} - \boldsymbol{\theta}_t) \right]. \tag{18}$$

We then add and subtract $\boldsymbol{\theta}_{\text{LLM}}$ in the last term above,

$$\boldsymbol{\theta}_{1:T} - \boldsymbol{\theta}_{\text{LLM}} \approx \sum_{t=1}^{T} \alpha_t(\boldsymbol{\theta}_t - \boldsymbol{\theta}_{\text{LLM}}) - \sum_{t=1}^{T} \alpha_t \mathbf{H}_0^{-1} \left[ \mathbf{H}_t(\boldsymbol{\theta}_{1:T} - \boldsymbol{\theta}_{\text{LLM}}) - \mathbf{H}_t(\boldsymbol{\theta}_t - \boldsymbol{\theta}_{\text{LLM}}) \right], \tag{19}$$

and multiply the whole expression by $\mathbf{H}_0$ and rearrange it to get the second expression in Eq. 18,

$$\begin{aligned} \left( \mathbf{H}_0 + \sum_{t=1}^{T} \alpha_t \mathbf{H}_t \right)(\boldsymbol{\theta}_{1:T} - \boldsymbol{\theta}_{\text{LLM}}) &\approx \sum_{t=1}^{T} \alpha_t \mathbf{H}_0(\boldsymbol{\theta}_t - \boldsymbol{\theta}_{\text{LLM}}) + \sum_{t=1}^{T} \alpha_t \mathbf{H}_t(\boldsymbol{\theta}_t - \boldsymbol{\theta}_{\text{LLM}}) \\ &= \sum_{t=1}^{T} \alpha_t(\mathbf{H}_0 + \mathbf{H}_t)(\boldsymbol{\theta}_t - \boldsymbol{\theta}_{\text{LLM}}). \end{aligned} \tag{20}$$

Multiplying the equation by inverse of $\mathbf{H}_0 + \sum_{t=1}^{T} \alpha_t \mathbf{H}_t$ and taking $\boldsymbol{\theta}_{\text{LLM}}$ to the right hand side gives us Eq. 12.

## A.3 Derivation of Model Merging as MAP of Bayes' Posterior

We will now connect our approach to Bayes' rule for linear regression. In this case, Eq. 10 can be rearranged to write as follows, where in the second term we have added and subtracted $\boldsymbol{\theta}_{1:T}$,

$$0 = -\mathbf{H}_0(\boldsymbol{\theta}_{1:T} - \boldsymbol{\theta}_{\text{LLM}}) + \sum_{t=1}^{T} \alpha_t \mathbf{H}_0(\boldsymbol{\theta}_t - \boldsymbol{\theta}_{1:T} + \boldsymbol{\theta}_{1:T} - \boldsymbol{\theta}_{\text{LLM}}) - \sum_{t=1}^{T} \alpha_t \mathbf{H}_t(\boldsymbol{\theta}_{1:T} - \boldsymbol{\theta}_t).$$

This equation is a stationarity condition of the following optimization problem,

$$\boldsymbol{\theta}_{1:T} = \arg\min_{\boldsymbol{\theta}} \left( 1 - \sum_{t=1}^{T} \alpha_t \right) \underbrace{\left[ -\tfrac{1}{2} \|\boldsymbol{\theta} - \boldsymbol{\theta}_{\text{LLM}}\|_{\mathbf{H}_0}^2 \right]}_{=\log p(\boldsymbol{\theta})} + \sum_{t=1}^{T} \alpha_t \underbrace{\left( -\tfrac{1}{2} \|\boldsymbol{\theta} - \boldsymbol{\theta}_t\|_{\mathbf{H}_0 + \mathbf{H}_t}^2 \right)}_{=\log p(\boldsymbol{\theta}|\mathcal{D}_t)}.$$

where we identify the prior to be $p(\boldsymbol{\theta}) = \mathcal{N}(\boldsymbol{\theta}|\boldsymbol{\theta}_{\text{LLM}}, \mathbf{H}_0^{-1})$, and the posterior on $\mathcal{D}_t$ to be $p(\boldsymbol{\theta}|\mathcal{D}_t) = \mathcal{N}(\boldsymbol{\theta}|\boldsymbol{\theta}_t, (\mathbf{H}_0 + \mathbf{H}_t)^{-1})$. We can therefore show that the stationarity condition corresponds to a maximum-a-posterior estimate of $p(\boldsymbol{\theta}|\mathcal{D}_{1:T})$ as follows,

$$p(\boldsymbol{\theta}|\mathcal{D}_{1:T}) \propto p(\boldsymbol{\theta}) \prod_{t=1}^{D} p(\mathcal{D}_t|\boldsymbol{\theta})^{\alpha_t} = p(\boldsymbol{\theta}) \prod_{t=1}^{D} \left[ \frac{p(\boldsymbol{\theta}|\mathcal{D}_t)}{p(\boldsymbol{\theta})} \right]^{\alpha_t} = p(\boldsymbol{\theta})^{1-\sum_{t=1}^{T} \alpha_t} \prod_{t=1}^{T} p(\boldsymbol{\theta}|\mathcal{D}_t)^{\alpha_t},$$

where log of the last term is equivalent to the objective function.

## A.4 Derivation of Data Removal

Our target model is the following model trained using Eq. 7 but without using $\mathcal{D}_t$,

$$\boldsymbol{\theta}_{\backslash t} = \arg\min_{\boldsymbol{\theta}} \ \bar{\ell}_{\backslash t}(\boldsymbol{\theta}) + \frac{\delta}{2}\|\boldsymbol{\theta}\|^2, \quad \text{where } \bar{\ell}_{\backslash t}(\boldsymbol{\theta}) = \sum_{i \in \{\mathcal{D}_{\text{Large}} \backslash \mathcal{D}_t\}} \ell_i(\boldsymbol{\theta}). \tag{21}$$

The LLM objective can then be written in terms of this objective:

$$\boldsymbol{\theta}_{\text{LLM}} = \arg\min_{\boldsymbol{\theta}} \ \bar{\ell}_{\backslash t}(\boldsymbol{\theta}) + \alpha_t \bar{\ell}_t(\boldsymbol{\theta}) + \frac{\delta}{2}\|\boldsymbol{\theta}\|^2, \tag{22}$$

where we assume that $\bar{\ell}_t$ is multiplied by a constant $\alpha_t$ in the original model.

As before, we can write the stationary conditions of $\boldsymbol{\theta}_{\text{LLM}}$, $\boldsymbol{\theta}_t$, and $\boldsymbol{\theta}_{\backslash t}$, respectively:

$$\delta\boldsymbol{\theta}_{\text{LLM}} = -\nabla\bar{\ell}_{\backslash t}(\boldsymbol{\theta}_{\text{LLM}}) - \alpha_t\nabla\bar{\ell}_t(\boldsymbol{\theta}_{\text{LLM}}),$$
$$\mathbf{H}_0(\boldsymbol{\theta}_t - \boldsymbol{\theta}_{\text{LLM}}) = -\nabla\bar{\ell}_t(\boldsymbol{\theta}_t), \tag{23}$$
$$\delta\boldsymbol{\theta}_{\backslash t} = -\nabla\bar{\ell}_{\backslash t}(\boldsymbol{\theta}_{\backslash t}).$$

Because our goal is to analyze $\boldsymbol{\theta}_{\backslash t} - \alpha_t(\boldsymbol{\theta}_{\text{LLM}} - \boldsymbol{\theta}_t)$, we multiply the second equation by $\alpha_t$, subtract it from the first equation, and then subtract the resultant from the third equation to get, the following,

$$\delta(\boldsymbol{\theta}_{\backslash t} - \boldsymbol{\theta}_{\text{LLM}}) + \alpha_t\mathbf{H}_0(\boldsymbol{\theta}_t - \boldsymbol{\theta}_{\text{LLM}}) = -\left[\nabla\bar{\ell}_{\backslash t}(\boldsymbol{\theta}_{\backslash t}) - \nabla\bar{\ell}_{\backslash t}(\boldsymbol{\theta}_{\text{LLM}})\right] + \alpha_t\left[\nabla\bar{\ell}_t(\boldsymbol{\theta}_{\text{LLM}}) - \nabla\bar{\ell}_t(\boldsymbol{\theta}_t)\right]. \tag{24}$$

We can now use Taylor's approximation to reduce gradient matching,

$$\nabla\bar{\ell}_{\backslash t}(\boldsymbol{\theta}_{\backslash t}) \approx \nabla\bar{\ell}_{\backslash t}(\boldsymbol{\theta}_{\text{LLM}}) + \nabla^2\bar{\ell}_{\backslash t}(\boldsymbol{\theta}_{\text{LLM}})(\boldsymbol{\theta}_{\backslash t} - \boldsymbol{\theta}_{\text{LLM}}).$$

For the second gradient term, we do not need to use the Taylor's approximation because it does not depend on $\boldsymbol{\theta}_{\backslash t}$, but our goal is to improve over task arithmetic, so we do it to derive a preconditioner,

$$\nabla\bar{\ell}_t(\boldsymbol{\theta}_{\text{LLM}}) \approx \nabla\bar{\ell}_t(\boldsymbol{\theta}_t) + \mathbf{H}_t(\boldsymbol{\theta}_{\text{LLM}} - \boldsymbol{\theta}_t). \tag{25}$$

Note that it is also possible to do the Taylor's approximation not around $\boldsymbol{\theta}_t$ but $\boldsymbol{\theta}_{\text{LLM}}$. Plugging these in Eq. 24, we can write,

$$\delta(\boldsymbol{\theta}_{\backslash t} - \boldsymbol{\theta}_{\text{LLM}}) + \alpha_t\mathbf{H}_0(\boldsymbol{\theta}_t - \boldsymbol{\theta}_{\text{LLM}}) = -\nabla^2\bar{\ell}_{\backslash t}(\boldsymbol{\theta}_{\text{LLM}})(\boldsymbol{\theta}_{\backslash t} - \boldsymbol{\theta}_{\text{LLM}}) + \alpha_t\left[\mathbf{H}_t(\boldsymbol{\theta}_{\text{LLM}} - \boldsymbol{\theta}_t)\right]$$
$$\implies \left[\delta\mathbf{I} + \nabla^2\bar{\ell}_{\backslash t}(\boldsymbol{\theta}_{\text{LLM}})\right](\boldsymbol{\theta}_{\backslash t} - \boldsymbol{\theta}_{\text{LLM}}) = -\alpha_t\left(\mathbf{H}_0 + \mathbf{H}_t\right)(\boldsymbol{\theta}_t - \boldsymbol{\theta}_{\text{LLM}})$$
$$\implies \boldsymbol{\theta}_{\backslash t} = \boldsymbol{\theta}_{\text{LLM}} - \alpha_t\left[\delta\mathbf{I} + \nabla^2\bar{\ell}_{\backslash t}(\boldsymbol{\theta}_{\text{LLM}})\right]^{-1}\left(\mathbf{H}_0 + \mathbf{H}_t\right)(\boldsymbol{\theta}_t - \boldsymbol{\theta}_{\text{LLM}})$$

which gives us the desired update given in Eq. 13.

## A.5 PROOF THAT OUR UPDATE FOR DATA-REMOVAL IS EXACT FOR LINEAR REGRESSION

The task removal update derived above is closely related to previous works on data removal. For instance, for linear model, our update recovers the popular influence function. We will now show this. Consider a large linear model (coincidentally also abbreviated as LLM) with full data $\mathcal{D} = (\mathbf{X}, \mathbf{y})$ where $\mathbf{y}$ is a vector of outputs and $\mathbf{X}$ is a matrix containing each feature vector as a row. The loss is $\bar{\ell}_{\text{LLM}}(\boldsymbol{\theta}) = \frac{1}{2}\|\mathbf{y} - \mathbf{X}\boldsymbol{\theta}\|^2$. Now, suppose we want to remove $\mathcal{D}_t = (\mathbf{X}_t, \mathbf{y}_t)$ from it. Then, we have a closed form solution for the full model and the model with removed data,

$$\boldsymbol{\theta}_{\text{LLM}} = \bar{\mathbf{H}}^{-1}\mathbf{X}^\top\mathbf{y}, \qquad \boldsymbol{\theta}_{\backslash t} = \bar{\mathbf{H}}_{\backslash t}^{-1}\mathbf{X}^\top\mathbf{y},$$

where $\bar{\mathbf{H}} = \nabla^2\left[\frac{1}{2}\|\mathbf{y} - \mathbf{X}\boldsymbol{\theta}\|^2 + \frac{1}{2}\|\boldsymbol{\theta}\|^2\right] = \mathbf{X}^\top\mathbf{X} + \delta\mathbf{I}$, and similarly $\bar{\mathbf{H}}_{\backslash t} = \mathbf{X}_{\backslash t}^\top\mathbf{X}_{\backslash t} + \delta\mathbf{I}$. A well known result in the influence function literature Cook (1977) is that the two quantities are related as

$$\boldsymbol{\theta}_{\backslash t} - \boldsymbol{\theta}_{\text{LLM}} = \bar{\mathbf{H}}_{\backslash t}^{-1}\mathbf{X}_t^\top(\mathbf{X}_t\boldsymbol{\theta}_{\text{LLM}} - \mathbf{y}_t). \tag{26}$$

We will now show that our update in Eq. 13 reduces to this for linear models.

We start with an expression for $\boldsymbol{\theta}_t$ trained using Eq. 8 but with the loss $\bar{\ell}_t(\boldsymbol{\theta}) = \frac{1}{2}\|\mathbf{y}_t - \mathbf{X}_t\boldsymbol{\theta}\|^2$. Using the second equation in the optimality condition of Eq. 23, we can write:

$$\mathbf{H}_0(\boldsymbol{\theta}_t - \boldsymbol{\theta}_{\text{LLM}}) = \mathbf{X}_t^\top(\mathbf{y}_t - \mathbf{X}_t\boldsymbol{\theta}_t) \qquad \implies \qquad (\mathbf{H}_0 + \mathbf{H}_t)\boldsymbol{\theta}_t = \mathbf{H}_0\boldsymbol{\theta}_{\text{LLM}} + \mathbf{X}_t^\top\mathbf{y}_t$$

where we use the fact that for linear models $\mathbf{H}_t = \mathbf{X}_t^\top\mathbf{X}_t$. We now simplify our update of Eq. 13 with $\alpha_t = 1$ where we use the above relationship in the third line below,

$$\begin{aligned}
\hat{\boldsymbol{\theta}}_{\backslash t} &= \boldsymbol{\theta}_{\text{LLM}} - \bar{\mathbf{H}}_{\backslash t}^{-1}\left(\mathbf{H}_0 + \mathbf{H}_t\right)(\boldsymbol{\theta}_t - \boldsymbol{\theta}_{\text{LLM}}) \\
&= \boldsymbol{\theta}_{\text{LLM}} - \bar{\mathbf{H}}_{\backslash t}^{-1}\left[(\mathbf{H}_0 + \mathbf{H}_t)\boldsymbol{\theta}_t - (\mathbf{H}_0 + \mathbf{H}_t)\boldsymbol{\theta}_{\text{LLM}}\right] \\
&= \boldsymbol{\theta}_{\text{LLM}} - \bar{\mathbf{H}}_{\backslash t}^{-1}\left(\mathbf{H}_0\boldsymbol{\theta}_{\text{LLM}} + \mathbf{X}_t^\top\mathbf{y}_t - (\mathbf{H}_0 + \mathbf{H}_t)\boldsymbol{\theta}_{\text{LLM}}\right) \\
&= \boldsymbol{\theta}_{\text{LLM}} - \bar{\mathbf{H}}_{\backslash t}^{-1}\left(\mathbf{X}_t^\top\mathbf{y}_t - \mathbf{H}_t\boldsymbol{\theta}_{\text{LLM}}\right) \\
&= \boldsymbol{\theta}_{\text{LLM}} - \bar{\mathbf{H}}_{\backslash t}^{-1}\left(\mathbf{X}_t^\top\mathbf{y}_t - \mathbf{X}_t^\top\mathbf{X}_t\boldsymbol{\theta}_{\text{LLM}}\right) \\
&= \boldsymbol{\theta}_{\text{LLM}} + \bar{\mathbf{H}}_{\backslash t}^{-1}\mathbf{X}_t^\top\left(\mathbf{X}_t\boldsymbol{\theta}_{\text{LLM}} - \mathbf{y}_t\right).
\end{aligned} \tag{27}$$

Therefore, our update reduces to Eq. 26.

A generalization of Eq. 26 to neural network is considered in Koh & Liang (2017) for the case of one-example removal. Their approach when applied to remove multiple examples at once redues to

$$\hat{\boldsymbol{\theta}}_{\backslash t} = \boldsymbol{\theta}_{\text{LLM}} + \bar{\mathbf{H}}_{\backslash t}^{-1} \mathbf{g}_t,$$

where $\mathbf{g}_t = \nabla \bar{\ell}_t(\boldsymbol{\theta}_{\text{LLM}})$. Our approach also recovers this result if we do not use the second Taylor's approximation for the second gradient matching term in Eq. 24. Essentially, this removes the contribution of the fine-tuned model and the steps are completely based on $\boldsymbol{\theta}_{\text{LLM}}$. It is not clear which approach is better but in practice it may depend on the fine-tune process which by doing multiple gradient steps may be able to get more information than a single gradient step $\mathbf{g}_t$. We hope to explore this point in a future study.

## B  PRACTICAL CONSIDERATIONS

### B.1  CHOICE OF LOSS FUNCTION & REGULARIZER

One design decision a practitioner has to take is the amount of regularization and the regularizer itself. While the presented derivations rely on weight decay and a quadratic regularizer, this restriction needs not to be made and is rather for simplicity of derivation. Consequently, we have found that even training models without a quadratic penalty gives similarly good results when merging with our objective. This might also be connected to the fact that early stopping of neural network training already implicitly regularizes the model towards its initialization, and not optimizing for too long will keep the fine-tuned model close. The $\delta$ chosen in weight decay is also included in Eq. 13 where $\mathbf{I} + \bar{\mathbf{H}}$ turns to $\delta \mathbf{I} + \hat{\mathbf{H}}_0$ if another value but $1/2$ is chosen and in Eq. 12 if $\mathbf{H}_0 = \mathbf{H}_0' + \delta \mathbf{I}$. Practically, a small $\delta > 0$ similar to what was outlined in Sec. 4.3 works well and $\delta = 0$ can become unstable, because the squared gradient approximation may also be 0 for some parameters.

The scaling of the Hessian approximation is also determined by the loss chosen for training. In case $\bar{\ell}$ is an average, the Fisher should also be averaged. This implies that all tasks are viewed as equally important. A different choice is to not average $\bar{\ell}$ and consequently the Fisher estimate is just summed. This carries a Bayesian interpretation of weighting tasks with more data higher, because observing more data means being more certain. Ultimately though which method performs better on test data can not be answered a priori and is a choice to make by the practitioner.

## C  EXPERIMENTAL DETAILS

This section contains experimental details to reproduce our study. We will also provide a repository containing the implementation upon acceptance.

### C.1  TASK ADDITION FOR CV

We use the implementation provided by (Ilharco et al., 2023) as a basis for our experiments. We use the ViT-B-32 model available at `https://github.com/mlfoundations/open_clip` as part of Open CLIP (Ilharco et al., 2021) and initialize all task models from it without training on a task beforehand. All models are trained using a large batch size of 128 on NVIDIA GPUs. We train the models using a modified version of the Adam optimizer (Kingma & Ba, 2015) that uses a decoupled quadratic penalty. We have found that a smaller number of epochs performs better, especially for task arithmetic but also for our method. Potentially, training long increases gradient mismatch and makes the fine-tuned models deviate far from the pretrained model. Therefore, we train for 5 epochs for small datasets and 10 for larger ones. We use a learning rate of $1e - 3$, $\beta_1 = 0.9$ and $\beta_2 = 0.999$. The Hessians of the task models are approximated using the squared gradient approximation on each task (normalized by data size), and for the pretrained model we use the identity multiplied by a scalar that is not tuned. We merge all models with the pretrained ViT and vary the scaling factor $\alpha_t$ which we keep same for each task model $\boldsymbol{\theta}_t$ for $t = 1, 2, \ldots, T$ within this experiment.

| Model | Hessian Approximation | IMDB | Yelp | RT | SST2 | Amazon | Avg. | True Avg. |
|---|---|---|---|---|---|---|---|---|
| RoBERTa | $\mathbf{H}_t = N \cdot \mathbf{I}$ | 94.6 | **97.3** | 89.5 | 93.3 | 96.6 | 94.3 | 96.5 |
| | Squared Gradient ($\mathbf{F}_{\text{sum}}$) | **94.7** (↑0.1) | **97.3** (-) | **90.2** (↑0.7) | **93.7** (↑0.4) | **96.7** (↑0.1) | **94.5** (↑0.2) | **96.6** (↑0.1) |

Table 5: The squared gradient approximation of the Hessian performs better than identity for task addition to an LLM for sentiment analysis, but even identity works well due to the improved scaling by accounting for gradient mismatch.

## C.2   TASK ADDITION FOR NLP

We use the RoBERTa-base checkpoint available on the huggingface hub Wolf et al. (2020) to initialize our model (available at `https://huggingface.co/roberta-base`) and also use all datasets in the versions that are available on the hub. We train each model for 2 epochs on IMDB and Yelp, 1 epoch on Amazon, and 5 epochs on the smaller SST2 and RottenTomatoes datasets. Furthermore, we subsample the training data for Yelp and Amazon and take the first 20% and 10% of the data to reduce computational load, and also because the datasets are much larger than the other three datasets and should not completely dominate them in the all-data baseline. We train all models using either AdamW Loshchilov & Hutter (2019) or a modified version of Adam (Kingma & Ba, 2015) with a quadratic penalty and use a learning rate of $1e - 5$ for training RoBERTa-base on IMDB and of $5e - 6$ for training the other models initialized from the IMDB model. We truncate the inputs at 384 tokens, and train using a batch size of 16 on NVIDIA GPUs. Furthermore, we set $\beta_1 = 0.9$ and $\beta_2 = 0.999$ as is standard in the transformers library, and use 100 warmup steps, as well as gradient norm clipping to unit norm. All squared gradient approximations are done by doing a single pass over the training data used to train each model, truncated to at most 100,000 examples. We then merge according to our proposed function by using a small $\delta = 1e - 10$. For Fisher averaging, we do not clip the Fishers at $1e - 6$ as is default in the implementation provided by Matena & Raffel (2022) but rather add the same $\delta$ as in our method. The "all-data" baseline is trained on the concatenation of all data used for the task-specific models for 2 epochs.

## C.3   TASK REMOVAL FOR NLP

For hallucination removal, we use the code provided by Daheim et al. (2023) to train a flanT5 model Longpre et al. (2023) on Wizard-of-Wikipedia Dinan et al. (2019). We use the same data augmentation of switching out the ground-truth document and train both the initial dialogue model and the model trained on the augmented data for 1 epoch using a batch size of 32. Again, we use the checkpoints provided on huggingface. We use AdamW or the modified version with a learning rate of $6 - 25e - 5$, $\beta_1 = 0.9$, and $\beta_2 = 0.999$. All Hessians are approximated by the squared gradient approximation of the Fisher by passing over the training data of each model once. We use $\delta = 1e - 10$ for merging and have found a smaller delta to generally provide better performance.

For toxicity removal, we tried to adhere to the setting of Ilharco et al. (2023) as closely as possible by employing the same prompts and datasets. In particular, we train GPT2 (Radford et al., 2019) or GPT-J (Wang & Komatsuzaki, 2021) on all data from Civil comments (Borkan et al., 2019) with a toxicity score of over $0.8$ and "subtract" this model from a pretrained GPT2 model (which we do not fine-tune on any data but use out-of-the-box given the checkpoint provided on the huggingface hub). The model is trained for only 1 epoch on the training data. We calculate the squared gradient approximation for GPT2 on the first 10000 instances of WikiText103 and for the task model on the task data. For evaluation we take the prompt from Ilharco et al. (2023), namely "I don't care if this is controversial." followed by a prompt from the RealToxictyPrompts dataset (Gehman et al., 2020). We use 'original' model available under `https://github.com/unitaryai/detoxify` from the Detoxify library (Hanu & Unitary team, 2020) to evaluate toxicity and score each output as toxic that exhibits a score greater than $0.2$.

# D ADDITIONAL RESULTS

## D.1 COMPARISON OF HESSIAN APPROXIMATIONS

We compare two different methods of setting the Hessians for the task addition experiment: 1) setting all Hessians to identity and 2) squared gradient approximation of the Fisher. In Table 5 we find identity Hessians to work surprisingly well, on par with Task Arithmetic and tuned $\alpha_t$, but still to be outperformed by the squared gradient approximation. This indicates that improved approximations could further improve merging performance which we leave to be explored in future work.

## D.2 SENTIMENT ANALYSIS WITH T5

We repeat the same task addition experiment for sentiment analysis as in the main paper Sec. 4.3 using T5 (Raffel et al., 2020) which is a pretrained encoder-decoder transformer (Vaswani et al., 2017) to outline that our method scales to different kinds of transformer (Vaswani et al., 2017) architectures. We use the pretrained T5 checkpoint without supervised fine-tuning, as SST2 is a part of that data, found under `https://huggingface.co/google/t5-v1_1-base`. We use a learning rate of $1e-4$ for the model trained on IMDB and $5e-5$ for the models trained on the other tasks which were initialized from it. The model predicts the sentiment of the text by generating the words "negative" or "positive". The results in Table 6 confirm that our method works across architectures.

| Model | Parametrization | IMDB | Yelp | RT | SST2 | Amazon | Avg. | True avg. |
|---|---|---|---|---|---|---|---|---|
| | | | | Accuracy (↑) | | | | |
| | All-data | 94.9 | 98.0 | 91.0 | 94.5 | 97.2 | 95.1 | 97.1 |
| T5$_{250M}$ | Task Arithmetic | 90.5 | 97.0 | 87.9 | 92.3 | 95.5 | 92.6 | 95.3 |
| | Ours | **93.6** (↑3.1) | **97.5** (↑0.5) | **90.7** (↑2.6) | **94.6** (↑2.3) | **96.5** (↑1.0) | **94.6** (↑2.0) | **96.4** (↑1.1) |

Table 6: For adding sentiment analysis tasks to T5 (Raffel et al., 2020) trained on IMDB, reducing gradient mismatch improves merging performance.

