# OpenReview forum: "Model Merging by Uncertainty-Based Gradient Matching"
_ICLR.cc/2024/Conference — ICLR 2024 poster_

### Official Review · Reviewer_1bWs · 2023-10-28

**Soundness:** 2 fair
**Presentation:** 3 good
**Contribution:** 3 good
**Rating:** 6
**Confidence:** 5

**Summary:**

The paper tries to theoretically understand the impact of gradient mismatch between tasks when merging these models together. The first shows how model merging and gradient mismatches are related to each other and shows the errors that are induced due to that, based on these insights they propose a new method to reduce gradient mismatch. Next, they demonstrate how many past model merging methods are the special case of their new method and finally establish the relationship with Bayesian inference. They conclude with a small set of experiments to demonstrate the usefulness of their method.

**Strengths:**

(S1) The originality of the work lies in providing the theoretical connection between model merging and the gradient mismatch problem (the identification of gradient mismatch as a problem of model merging cannot be attributed to this work, see weaknesses). Moreover, the connection with the Bayesian inference is also interesting. Additionally, the insight that the work extends RegMean (Jin et. al;2023) to non-linear parts of transformers is useful.

(S2) The paper is well-written for the most part and easy to follow along. However, there are things that are not clear and are listed as Questions below.

(S3) The work improves the understanding of model merging and might be useful to the people working on the model merging.

**Weaknesses:**

(W1) The idea of identifying gradient mismatch as a problem has been claimed as one of the main contributions throughout the paper (abstract, intro, and other sections). However past works like TIES-Merging [1] have identified this problem and proposed detailed empirical studies to quantify the degree of this problem and then propose some fixes that lead to significantly improved performance.

(W2) Moreover, in the current version of the paper, TIES-Merging is discussed in passing but the differences compared to that works are not properly highlighted.

(W3) The experimental section is pretty thin and the results presented are weak. See more details in the Questions below.

**Questions:**

**Need to address these questions for me to retain my score:**

Q1: The papers need to be positioned better, to adjust the main contribution and highlight the similarities/differences compared to TIES-Merging [1] which claims to identify and ameliorate interference when merging models.



**Need to address these as well for me to consider increasing my score:**

Q2: Ideally all the experiments should also compare with TIES-Merging as that method is the closest to the final method proposed in this paper. And addresses the exact same problem that this paper tries to get at. Hence, not including that as a baseline leaves lots of open questions about the utility of the proposed method. If these comparisons are added then I will update my score as I feel this work makes a good theoretical contribution but the experimental section leaves a lot of ambiguity about the utility of the final method.


Q3: The experimental results are very weak and seem insignificant. For example in Table (nlp), the experimental setting seems to be not well designed due to multiple reasons (i) the performance of all the methods lies between 96.1-96.8 which is quite a narrow range to make any claim about any of the methods performing better or worse than the others. for example, the difference between your method and TA is 0.3% which is not significant from my experience. Hence, either the experimental setting is too simple to highlight the differences between these methods or the methods all perform the same. An experimental setting from past papers like Task Arithmetic, TIES-Merging can be adopted for such experiments.

Q4: It is not clear how the approximations made in the paper about using fisher instead of hessian after the gradient mismatch as the model becomes bigger, something on this would be useful. Moreover, could these be the reasons why the method does not lead to significant improvements in both vision and NLP settings?


Q5: Figure-1 (right), it is not clear to me how this gradient mismatch is computed. Seems like you are adding 5 tasks on Roberta (IMDB) but then what circles represent the gradient mismatch between which models on which data? Is it a pairwise comparison of gradient mismatch or are the models being added? Please clarify exactly what this figure means.


Q6: Moreover, the Figure-1 (right) it is shown that as the gradient mismatch decreases the test error also decreases significantly (by ~2). however, this finding seems to be inconsistent with the results in table-3 where the performance difference between TA and your method is very minimal (0.3). What is the reason behind this? In general, what is the reason behind not leading to enough improvements over TA in the nlp setting even when there is a significant gradient mismatch for TA?


**Other questions on clarifications and details:**

Q7: How is the alpha selected in your proposed method? It is mentioned in many places that alpha is not tuned.

Q8: Please specify the number of samples you use to compute the fisher.

Q9: For Figure-2 (left) the best performance for both TA and your method seems to be comparable to each other. I agree that your method might not need to tune for alpha but in most practical cases obtaining a small validation set and tuning alpha is not that hard. Moreover, the proposed method need to compute the fisher (requires backward pass on a subset of training data) whereas TA need validation set to tune alpha (inference on a small number of val example), so overall the peak memory usage of the proposed method would be higher while TA requires additional data. This trade-off should be highlighted in the paper.

Overall, I feel that the theoretical contributions of this work are nice and would be useful to the community, I expect the author to at least position the contributions of the work better in light of past works. Moreover, strengthening the experiments section would highly increase the quality of this works

[1] Resolving interference when merging models, Yadav et. al.

---

> ### Author Response · Authors · 2023-11-17
> **Thank you for your review! (part 1)**
>
> We thank the reviewer for their in-depth review and appreciation for our method. The main criticism is lack of comparison to a recent method TIES-merging (accepted at NeurIPS 2023) and experimental results being weak. We disagree with both but, to address the reviewers’ concerns, we have added a discussion to TIES-merging and an additional comparison in Table 3.
>
>
> > Q1: The papers need to be positioned better, to adjust the main contribution and highlight the similarities/differences compared to TIES-Merging [1] which claims to identify and ameliorate interference when merging models.
>
> A1: Thanks for the suggestion. We have now added a few sentences to highlight the differences to TIES-Merging (see the last line in Page 2 and also Section 3.2.1 in the revised version). In summary, the method can be related to ours by using a binary mask (similarly to Ansell et al. 2022 in Table 1) but the paper [1] does not mention gradient matching and it remains unclear how the operations used in TIES directly lead to reductions in gradient mismatch.
> While gradient mismatch is calculated between each $\theta_t$ and the target model $\theta_{1:T}$, their concept of interference is outlined between the $\theta_t$.
>
>
> >Q2: Ideally all the experiments should also compare with TIES-Merging as that method is the closest to the final method proposed in this paper. And addresses the exact same problem that this paper tries to get at. Hence, not including that as a baseline leaves lots of open questions about the utility of the proposed method. If these comparisons are added then I will update my score as I feel this work makes a good theoretical contribution but the experimental section leaves a lot of ambiguity about the utility of the final method.
>
> A2: We request the reviewer to reconsider the need to compare *all* the experiments. The ICLR policy clearly advises to not have to compare with papers posted on arXiv after May 28, 2023. We do not agree that the absence of such comparisons “leaves a lot of ambiguity about the utility” of our method (see A3.1 below). We also disagree that TIES-Merging is “closest” to our method: our proposal is to reduce gradient mismatch which is different from the trimming used in TIES; gradient matching is never even mentioned in the TIES paper. We respectfully request the reviewer to reconsider their opinion.
>
> **To address the reviewers’ concerns, we have added a comparison in Table 3 which is feasible to do in this short time. We kindly ask the reviewer to reevaluate their rating and increase their score based on the new comparison.**
>
> >Q3.1: The experimental results are very weak and seem insignificant. For example in Table (nlp), the experimental setting seems to be not well designed due to multiple reasons (i) the performance of all the methods lies between 96.1-96.8 which is quite a narrow range to make any claim about any of the methods performing better or worse than the others. for example, the difference between your method and TA is 0.3% which is not significant from my experience. Hence, either the experimental setting is too simple to highlight the differences between these methods or the methods all perform the same.
>
> A3.1: We respectfully disagree: improvements (like 0.3%) are compared to a “well-tuned TA” method in Table 3, but without such tuning the baselines are much worse (also see this in Fig. 2). Larger gains with respect to other measures are also obtained in Table 2 and 4 (e.g., a gain of 3.4 for fluency PPL). A 0.3% increase is also not insignificant for a large test set (in this case containing close to 500,000 test examples). We understand that, when only focusing on a specific measure (test accuracy) with respect to one method (well-tuned TA), one may call the improvements marginal, but this does not mean that the presented methods are ineffective (and insignificant) in all respects. We request the reviewer to reconsider their opinion.
>
> >Q3.2: An experimental setting from past papers like Task Arithmetic, TIES-Merging can be adopted for such experiments.
>
> A3.2: Our experiments are already adopted from the task arithmetic papers (for example, Figure 2 (left) and Table 4 (left)) and other recent works (Table 4 (right)). TIES-Merging is not used because it is too recent but we will consider adding further experiments for the final version.
>
> >Q4: It is not clear how the approximations made in the paper about using fisher instead of hessian after the gradient mismatch as the model becomes bigger, something on this would be useful. Moreover, could these be the reasons why the method does not lead to significant improvements in both vision and NLP settings?
>
> A4: We expect better approximations to give better results but it is not correct to directly attribute bad performance to the approximation. We use the GGN approximation which is a popular choice and is effective even at large scale [2,3,4,5]. We plan to study the effect of this approximation in the future.
> (pt.2 below)

---

> > ### Author Response · Authors · 2023-11-17
> > **Author response part 2**
> >
> > >Q5: Figure-1 (right), it is not clear to me how this gradient mismatch is computed. Seems like you are adding 5 tasks on Roberta (IMDB) but then what circles represent the gradient mismatch between which models on which data? Is it a pairwise comparison of gradient mismatch or are the models being added? Please clarify exactly what this figure means.
> >
> > A5: It is a pairwise comparison of the gradient mismatch between the merged model and the target model (trained on all data). Each marker represents one task; see Section 4.3 on the details of the experimental setup.
> >
> >
> > >Q6 & Q7: Moreover, the Figure-1 (right) it is shown that as the gradient mismatch decreases the test error also decreases significantly (by ~2). however, this finding seems to be inconsistent with the results in table-3 where the performance difference between TA and your method is very minimal (0.3). What is the reason behind this? In general, what is the reason behind not leading to enough improvements over TA in the nlp setting even when there is a significant gradient mismatch for TA? …..How is the alpha selected in your proposed method? It is mentioned in many places that alpha is not tuned.
> >
> > A6 & A7: This is a misunderstanding; there is no such inconsistency. The reviewer is talking about the “well-tuned TA” baseline, but Fig. 1 uses default TA with $\alpha_t =1$ which (consistently) performs worse in Table 3 as well (e.g., 90.5% on IMDB compared to our 94.7% in the first column). In our method, we always set $\alpha_t = 1$ and do not use any tuning, yet obtain better results than a well-tuned TA baseline. We have now improved the writing of the paper to avoid the misunderstanding (Fig. 1, Table 2 and Sec. 4.3).
> >
> > >Q8: Please specify the number of samples you use to compute the fisher.
> >
> > A8: We use either the entire train dataset or at most the first 100,000 examples from it if the dataset is larger to reduce computational burden. We will add this to our paper (Appendix C.2).
> >
> >
> > >Q9: For Figure-2 (left) the best performance for both TA and your method seems to be comparable to each other. I agree that your method might not need to tune for alpha but in most practical cases obtaining a small validation set and tuning alpha is not that hard. Moreover, the proposed method need to compute the fisher (requires backward pass on a subset of training data) whereas TA need validation set to tune alpha (inference on a small number of val example), so overall the peak memory usage of the proposed method would be higher while TA requires additional data. This trade-off should be highlighted in the paper.
> >
> > A9: Thanks for the suggestion: we have now highlighted this point in the paper (see the last paragraph of Section 3.3). For a large number of tasks, setting alpha can be expensive, but we agree that for a small number of tasks it can be easy when a validation set is known and available. We also stress that we do not need to do a full pass through the data because Adam-style methods can be easily modified to obtain an estimate of the Fisher during training and without overhead (a common practice to obtain online Fisher in continual learning; cf. [6] or [2; Section 11.2]).
> >
> > **References**
> >
> > [1] P. Jadav et al, Resolving Interference When Merging Models, NeurIPS, 2023.
> >
> > [2] J. Martens. New insights and perspectives on the natural gradient method. The Journal of Machine Learning Research 21.1 (2020): 5776-5851.
> >
> > [3] N. N. Schraudolph, Fast curvature matrix-vector products for second-order gradient descent.
> > Neural Computation, 14(7):1723–1738, 2002
> >
> > [4] A. Graves, Practical variational inference for neural networks, NeurIPS, 2011.
> >
> > [5] M. Khan et al., Fast and scalable bayesian deep learning by weight-perturbation in adam. In International conference on machine learning (pp. 2611-2620), PMLR, 2018.
> >
> > [6] J. Schwarz et al. Progress & compress: A scalable framework for continual learning. International conference on machine learning. PMLR, 2018.

---

> > > ### Comment · Reviewer_1bWs · 2023-11-21
> > >
> > > Thank you for providing a detailed response. I will keep my original score because I still feel that the main experimental setting for the main merging experiments is too trivial, most methods perform similarly to each other, hence the setting is not convincing enough for me. However, I really love the theoretical insights and hence still believe that this paper adds a lot of value to the community.
> > >
> > > A6 & A7: From Figure2 it seems like the value of alpha is being tuned. Second, if you are using alpha=1 then the performance would be worse than TA in the vision setting, I am still not sure about alpha selection and at all places in the paper, this should be clarified, for example for figure 2.
> > >
> > > "We also disagree that TIES-Merging is “closest” to our method: our proposal is to reduce gradient mismatch which is different from the trimming used in TIES; gradient matching is never even mentioned in the TIES paper. We respectfully request the reviewer to reconsider their opinion." -> They show interference for task vectors ($\tau = \theta_{ft} - \theta_{init}$), which is the accumulated gradient from all the training steps, hence the interference they show is between accumulated gradient. However, I understand the work might be too recent which is precisely why my initial rating accounted for this and was positive. My score was provided independent of this comparison. Regardless of that, I agree that this is the first work that tries to theoretically understand the interference/gradient mismatch problem and highlight very neat connections, however not the first to identify it the problem.
> > >
> > > Reply to A8: Ideally it would be good to see if fisher can be approximated using a few samples say less than 1000. I think that might work. This is a suggestion and results on this are not needed right now.

---

> > > > ### Author Response · Authors · 2023-11-23
> > > > **Thank you for the follow-up!**
> > > >
> > > > We are glad to hear that the reviewer believes that our theoretical insights add a lot of value to the community.
> > > > Below we add some further clarifications:
> > > >
> > > > > A6 & A7: From Figure2 it seems like the value of alpha is being tuned. Second, if you are using alpha=1 then the performance would be worse than TA in the vision setting, I am still not sure about alpha selection and at all places in the paper, this should be clarified, for example for figure 2.
> > > >
> > > > No it is not tuned and we will clarify this in the paper. The plot simply shows results for different alpha on the test data, not on a validation set. In our method, alpha is always set to 1.0 during training.
> > > >
> > > > > They show interference for task vectors ($\tau = \theta_{ft} - \theta_{init}$), which is the accumulated gradient from all the training steps, hence the interference they show is between accumulated gradient.
> > > >
> > > > We stress the “gradient interference” is different from “gradient mismatch”. For mismatch the target model is considered, which is not the case for interference (which as you said is about accumulated gradient).
> > > >
> > > > > I will keep my original score because I still feel that the main experimental setting for the main merging experiments is too trivial, most methods perform similarly to each other, hence the setting is not convincing enough for me.
> > > >
> > > > We request you to not judge the quality of our experiments by focusing on on one table, because we do get large improvements in other experiments, for example for data removal in Table 4.
> > > > Please note that such marginal improvements are also reported in existing works too, for instance, Table 3 in the TIES-Merging paper [1], and Table 1 in the RegMean paper [2].
> > > >
> > > > > However, I understand [TIES-Merging] might be too recent... My score was provided independent of this comparison. Regardless of that, I agree that this is the first work that tries to theoretically understand the interference/gradient mismatch problem and highlight very neat connections....
> > > >
> > > > Thanks for the positive comments. We hope that our explanations help to shed light on some of the confusion. We still hope that you will consider raising the score.
> > > >
> > > > *References*
> > > >
> > > > [1] P. Jadav et al, Resolving Interference When Merging Models, NeurIPS, 2023.
> > > >
> > > > [2] X. Jin et al., Dataless Knowledge Fusion by Merging Weights of Language Models, ICLR, 2023.

---

### Official Review · Reviewer_V4z1 · 2023-10-29

**Soundness:** 2 fair
**Presentation:** 2 fair
**Contribution:** 3 good
**Rating:** 6
**Confidence:** 2

**Summary:**

The describes a structured way of understanding linear combination of parameters. The concept of ``target model`` is introduced as a way of measuring fitness of merges. Subsequently, modification of the ``Task arithmetic`` loss is introduced such that the gradient mismatch between the target model and the averaged models is minimized. Experimental results show comparable results.

**Strengths:**

- The introduction of the paper is very well written and framed the problem in clearly.
- The idea of defining a ``target model`` is very useful concept in this space.

**Weaknesses:**

- Section 3, overall, was difficult to follow with seemingly several notational errors and cluttered paragraphs, see questions and suggestions section.

**Questions:**

**Questions**

- Eq(3)  should be
$$\alpha_{1}\bar{\ell}_{1}(\theta) + \alpha_{2}\bar{\ell}_{2}(\theta)$$
 right?
In other words, the optimization is looking for $\theta$ that optimizes both losses which is $\theta_{1+2}$. If this is not a mistake then Eq(5) is wrong.
- what does $t$ stand for in Eq(8)
- the error between $\bar{\theta}_{TA}$ and $\theta_{1:T} = \theta_{1:T}$? please clarify/correct?

**Suggestions**
- The discussion in the last paragraph in page 3 is best to be had in the experimental section with some data.
   or under its own section with further details.
- Section 3.1 is difficult to follow/understand, mainly because of several math annotation issue, and not well
   organized paragraphs. For instance, the first two paragraphs can simply be phrased as `target model` definition
   rather than using unnecessary details and confusing notations.
- I generally, like the framing of the problem and the idea of ``target model``. I think the paper has good potential. I suggest re-writing of section 3, highlighting the problem and the solution (perhaps computational aspect and other details) and differing questions of generality and applications to later section.

---

> ### Author Response · Authors · 2023-11-17
> **Thank you for your review!**
>
> We thank the reviewer for their review, appreciation of the usefulness of our method, and for pointing out various typos. Their suggestions have helped us greatly in improving the writing of Section 3. We hope that the reviewer would consider increasing their score if this answers their concerns. Below we address all questions:
>
>
> > R3’s suggestions on improving the presentation in Section 3.
>
> A: Thank you for the constructive comments! We have now rewritten the initial paragraphs of Section 3.1 to clarify the model definitions and split Section 3.2 into further subsections to ease readability. We also have fixed the notation mistakes, thanks again for pointing these out. For the final version, we will make sure to take your suggestions into account!
>
>
> > The discussion in the last paragraph in page 3 is best to be had in the experimental section with some data. or under its own section with further details.
>
> A: Thank you for the suggestion. For now we have kept this paragraph at the end of Section 3 but we will add an additional experiment for the final version and add further discussion. Our experiments already outline the claim at the end of the paragraph: we can use gradient mismatch to improve model merging by reducing it, for example, in Figure 1 (right) or Table 2.
>
>
> > Q1: Eq(3) should be $$\alpha_{1}\bar{\ell_{1}}(\theta) + \alpha_{2}\bar{\ell_{2}}(\theta)$$ right? In other words, the optimization is looking for $\theta$ that optimizes both losses which is $\theta_{1+2}$. If this is not a mistake then Eq(5) is wrong.
>
> A: Yes, thank you for the careful reading. We have corrected it!
>
> >Q2: what does $t$ stand for in Eq (8)?
>
> A: The $t$ is arbitrary but fixed and stands for a specific task or dataset $\mathcal{D}_t$ that the model $\theta_t$ is trained on. During merging we combine all these $\theta_t$, of which there are $T$ in total. We have made this clearer in Section 3.1.
>
>
> > Q3: What is the error between $\theta_{TA} \text{ and } \theta_{1:T} = \theta_{1:T}$? Please clarify/correct?
>
> A: The error is calculated between $\theta_{1:T}$ and $\bar{\theta_{TA}}$.
> The former is the target model, and the latter its approximation through Task Arithmetic.
> We can rewrite Eq. (10) to make this more explicit by subtracting the left summand on the RHS (which is $\bar{\theta_{TA}}$).
> Then, we obtain the following equation:
> $\theta_{1:T} - \bar{\theta_{TA}} = -\sum_{t=1}^T \alpha_t H_0^{-1}[ \nabla \bar{l_t}(\theta_{1:T}) - \nabla \bar{l_t}(\theta_t)]$.
> On the right we therefore again find the gradient mismatch as the error. We have made this clearer in the paper and rewritten the equation, thank you for the comment!
>
> > R3: Experimental results show comparable results.
>
> A: We would like to emphasize that we get consistent improvements both in terms of performance over a well-tuned task arithmetic baseline, with scaling factors determined on the test data, and better robustness to choice of scale. This was the goal of reducing gradient mismatch. Furthermore, we would like to point out the improvements for data removal shown in Table 4, which are consistent across different tasks and models.

---

> > ### Author Response · Authors · 2023-11-22
> > **Gentle reminder.**
> >
> > Dear Reviewer, since the discussion period will be closing soon, please let us know if there are any further questions.

---

> > > ### Comment · Reviewer_V4z1 · 2023-12-03
> > > **Thank you for the response.**
> > >
> > > Thank you for taking the time to respond to my comments. I went through the modified version of the paper and other reviewers' feedback. My primary concern with the paper was readability, in particular section 3. It has improved from the original version! I elevated my score to 6.

---

### Official Review · Reviewer_vWCx · 2023-10-29

**Soundness:** 4 excellent
**Presentation:** 4 excellent
**Contribution:** 4 excellent
**Rating:** 6
**Confidence:** 4

**Summary:**

The paper addresses an interesting problem in the domain of model merging and offers a novel perspective by connecting gradient mismatches to the inaccuracy of weighted-averaging methods. The paper also proposes a new uncertainty-based scheme to improve model merging, which is a valuable contribution.

**Strengths:**

+ The authors connect the inaccuracy of weighted-averaging to mismatches in the gradients and propose a new uncertainty-based scheme to improve performance by reducing the mismatch.
+ The authors propose a unified explanation on previous model merging technique.
+ The new method shows consistent improvements for large language models and vision transformers in terms of performance and robustness to hyperparameters.

**Weaknesses:**

+ My major concern lies in the problem setup. I admit that model merging is a well-defined problem with much previous literature, as is discussed in the submission. But I still wonder why we need this technology. If we could obtain the data for each task, why don't we simply perform multi-task learning on these data? If we couldn't, how could we obtain the fisher information matrix on each task, which is required to approach Eq.12? It seems like a contradiction and I think more clarification on the application scenario of the model merging technique is needed, in spite of the abundance of previous literature.

+ The second concern is an important missing baseline. The derivation in section 3 is similar in some degree to Regmean[1] though the latter takes linear regression as an example and then extrapolates to neural networks.  Therefore,  I would list Regmean as one of the must-to-compare baseline methods.

**Questions:**

+ In my opinion, the model merging technique takes two or more models as input and outputs a merged model. Therefore, the performance of a merged model on down-stream tasks (compared to the unmerged model) is only a single datapoint in the experiment. In other words, what if we use different hyper-parameters to train the base model on each task? Will your method outperform others under other hyper-parameters?

---

> ### Author Response · Authors · 2023-11-17
> **Thank you for your review!**
>
> We thank the reviewer for their review and their appreciation for the value of our contribution.
> Below we answer all questions:
>
>
> >Q1: My major concern lies in the problem setup. I admit that model merging is a well-defined problem with much previous literature, as is discussed in the submission. But I still wonder why we need this technology. If we could obtain the data for each task, why don't we simply perform multi-task learning on these data?
>
> A1: In summary, finetuning can be highly resource-intensive, especially for large models, while model merging allows us to reuse already trained models. Finetuning on a new task can also lead to forgetting older tasks and retraining can be very costly at times, especially for removing data from a large pretrained corpus. To clarify, we have added a note on this in the section on data removal (Section 3.2.2); also see our response to Q1 of Reviewer idAX.
>
>
> >Q2:  If we couldn't, how could we obtain the fisher information matrix on each task, which is required to approach Eq.12? It seems like a contradiction [...]
>
> A2: Many optimizers, such as **Adam, already estimate the diagonal Fisher** which could be made available with the model, see [1; Section 11.2] for a discussion on how Adam approximates diagonal Fisher. Then, there is no overhead and if this estimate is shared, data can remain private. Thank you for pointing this out, we have emphasized this more in our discussion at the end of Section 3.3.
>
>
> >Q3: The second concern is an important missing baseline. The derivation in section 3 is similar in some degree to Regmean[1] though the latter takes linear regression as an example and then extrapolates to neural networks. Therefore, I would list Regmean as one of the must-to-compare baseline methods.
>
> A3: Thank you for the constructive suggestion, we have added RegMean to Table 3! In short, RegMean shows competitive performance to other well-tuned baselines and our method.
>
>
> > Q4: In my opinion, the model merging technique takes two or more models as input and outputs a merged model. Therefore, the performance of a merged model on down-stream tasks (compared to the unmerged model) is only a single datapoint in the experiment. In other words, what if we use different hyper-parameters to train the base model on each task? Will your method outperform others under other hyper-parameters?
>
> A4: For different hyperparameters, the performance will likely further improve. For our method, **we do not tune the hyperparameters** and use the same learning rate used in [2] for RoBERTa and the same default hyperparameters for AdamW taken from the transformers library [3].
>
> **References**
>
> [1] J. Martens. New insights and perspectives on the natural gradient method. The Journal of Machine Learning Research 21.1 (2020): 5776-5851.
>
> [2] Y. Liu, et al. Roberta: A robustly optimized bert pretraining approach. arXiv 1907.11692, 2019.
>
> [3] T. Wolf, et al. Transformers: State-of-the-art natural language processing. Proceedings of the conference on empirical methods in natural language processing: system demonstrations, 2020.

---

> > ### Author Response · Authors · 2023-11-22
> > **Gentle reminder.**
> >
> > Dear Reviewer, since the discussion period will be closing soon, please let us know if there are any further questions.

---

### Official Review · Reviewer_idAX · 2023-10-31

**Soundness:** 4 excellent
**Presentation:** 4 excellent
**Contribution:** 3 good
**Rating:** 6
**Confidence:** 3

**Summary:**

The paper introduces a powerful method to average two models. Specifically, the proposed method averages the model by minimizing the gradient mismatch of different models. The paper provides a deep analysis of why their method makes more sense than others.
 Also, the paper validates their method in multiple datasets from both the NLP domain and the image domain.

**Strengths:**

From my viewpoint, the weight of LLM is knowledge abstracted from data, which stresses the importance of quickly merging knowledge learned from the dataset. I believe the topics of the paper fit into this conference and have a certain inspiration for future works in this domain.

The motivation of this paper is extremely clear by analyzing the gradient of merged models. When I read the paper, I enjoyed the motivation despite the heavy math.

**Weaknesses:**

I have some minor concerns about this paper. Before I lay out the weaknesses list, I would like to mention that I’m not an expert on NLP and my comments are probably incorrect.

Finetuning vs data-driven model averaging. Maybe I don’t have the background.  I’m curious about the advantage of the proposed model merging over simply fine-tuning the model. In my understanding, for the proposed method to work, we would need data to calculate the gradient matrix -- that’s why I call the proposed method as a data-driven model averaging. In this case, why don’t we just simply fine-tune the averaged model using the LORA on the data in hand? And fine-tuning sounds more straightforward. Thus, I would recommend having a discussion/quick comparison between those two.

Again, I’m a bit concerned about the time efficiency since the proposed method requires the second-order Hessian matrix, especially when compared with the simple strategy. Although it doesn’t matter for the inference, it might be still worth knowing if this Hessian calculation is practical or not. So I suggest to make it clear.

In short, I have some concerns about the comparison with simple fine-tuning and time efficiency. So I currently vote for the weak accept. Again, it might be because I don’t have too much domain knowledge. So I would be happy to hear back from the authors during the rebuttal in case I misunderstand anything.

**Questions:**

Please address the question above.

---

> ### Author Response · Authors · 2023-11-17
> **Thank you for your review!**
>
> We thank the reviewer for their review and helpful suggestions. Below we address the questions they asked.
>
> > Q1: Finetuning vs data-driven model averaging. Maybe I don’t have the background. I’m curious about the advantage of the proposed model merging over simply fine-tuning the model. In my understanding, for the proposed method to work, we would need data to calculate the gradient matrix -- that’s why I call the proposed method as a data-driven model averaging. In this case, why don’t we just simply fine-tune the averaged model using the LORA on the data in hand? And fine-tuning sounds more straightforward. Thus, I would recommend having a discussion/quick comparison between those two.
>
> A1: Thank you for your suggestion! We already have a comparison to finetuning in Table 3. There, the “all-data” baseline is finetuned on all datasets at once.
> In general, finetuning can be resource-intensive, while model merging allows one to reuse already trained models. Finetuning on a new task can also lead to forgetting older tasks. Retraining is often costly, especially for removing small subsets of a large pretraining corpus from a model. Then, merging is much **more efficient than finetuning**. We have added further discussion of this in Section 3.2.2. We also stress that estimating Hessians does not require data, because we can do this in an online fashion during training (see our response to the next question Q2).
>
>
>
> > Q2: Again, I’m a bit concerned about the time efficiency since the proposed method requires the second-order Hessian matrix, especially when compared with the simple strategy. Although it doesn’t matter for the inference, it might be still worth knowing if this Hessian calculation is practical or not. So I suggest to make it clear.
>
> A2: **The Hessian calculation is practical**. Our squared gradient approximation to the Hessian only incurs a small overhead (a pass over a subset of the data is often enough) but it can also be obtained in an online fashion. For example, when running second-order optimizers (or approximations thereof such as Adam, see [1; Section 11.2]), an **approximation of the Hessian is calculated during training as a “by-product”**. See also our response Q9 to Reviewer 1bWs. We have emphasized this further in our paper (end of Section 3.3) and hope that this makes it clearer.
>
>
>
> **References**
>
> [1] J. Martens. New insights and perspectives on the natural gradient method. The Journal of Machine Learning Research 21.1 (2020): 5776-5851.

---

> > ### Author Response · Authors · 2023-11-22
> > **Gentle reminder.**
> >
> > Dear Reviewer, since the discussion period will be closing soon, please let us know if there are any further questions.

---

> > > ### Comment · Reviewer_idAX · 2023-11-22
> > > **Thanks for the feedbacks!**
> > >
> > > Thanks for authors' feedback. And sorry for the late. I think I have no further questions -- most of my questions have been addressed and I'll reconsider my rating along with my justification later. Thanks!

---

### Author Response · Authors · 2023-11-17
**Thank you for your reviews!**

We would like to thank the reviewers for their insightful comments.
We feel encouraged that the reviewers agree that our work adds theoretical understanding to model merging and can inspire future works.

The main concerns seem to be:

(1) The problem set-up of model merging

(2) The feasibility of Hessian approximations

(3) Comparison to & discussion of further methods (RegMean and TIES-merging)

We have addressed (1) and (2) sufficiently in our responses which emphasize the benefits of model merging and show that good estimates of the Hessians can be done during training without overhead.
We have also addressed (3) by adding further discussions and comparisons. We have also improved the writing and presentation, but will continue to do so taking the reviewers’ suggestions into account.

---

### Meta-Review · Area_Chair_dg8C · 2023-12-08

**Metareview:**

This paper connects model merging to preexisting notions of uncertainty estimation, specifically in terms of the gradient mismatch between different models to be merged. The paper contains quite a lot of analysis that ultimately leads to a new merging method whose primary benefit is increased robustness to the choice of hyperparameters. On the whole, the paper sheds some new light on the important problem of model merging. The empirical gains of the proposed method are not great, but this is offset by the analysis.

**Justification For Why Not Higher Score:**

While the analysis in this paper is interesting and useful, the ultimate outcome of the analysis does not provide a dramatically better algorithm.

**Justification For Why Not Lower Score:**

All reviewers recommended acceptance.

---

### Decision · Program_Chairs · 2024-01-16

Accept (poster)